# A deubiquitylase with an unusually high-affinity ubiquitin-binding domain from the scrub typhus pathogen *Orientia tsutsugamushi*

Jason M. Berk[1,5], Christopher Lim[1,5], Judith A. Ronau[1,3], Apala Chaudhuri[2], Hongli Chen[1], John F. Beckmann[1,4], J. Patrick Loria[1,2], Yong Xiong [1✉] & Mark Hochstrasser [1✉]

Ubiquitin mediated signaling contributes critically to host cell defenses during pathogen infection. Many pathogens manipulate the ubiquitin system to evade these defenses. Here we characterize a likely effector protein bearing a deubiquitylase (DUB) domain from the obligate intracellular bacterium *Orientia tsutsugamushi*, the causative agent of scrub typhus. The Ulp1-like DUB prefers ubiquitin substrates over ubiquitin-like proteins and efficiently cleaves polyubiquitin chains of three or more ubiquitins. The co-crystal structure of the DUB (OtDUB) domain with ubiquitin revealed three bound ubiquitins: one engages the S1 site, the second binds an S2 site contributing to chain specificity and the third binds a unique ubiquitin-binding domain (UBD). The UBD modulates OtDUB activity, undergoes a pronounced structural transition upon binding ubiquitin, and binds monoubiquitin with an unprecedented ~5 nM dissociation constant. The characterization and high-resolution structure determination of this enzyme should aid in its development as a drug target to counter *Orientia* infections.

---

[1] Department of Molecular Biophysics and Biochemistry, Yale University, New Haven, CT 06520, USA. [2] Department of Chemistry, Yale University, New Haven, CT 06520, USA. [3] Present address: Discovery, Research and Development, AbbVie, Inc., North Chicago, IL 60064, USA. [4] Present address: Department of Entomology and Plant Pathology, Auburn University, Auburn, AL 36830, USA. [5] These authors contributed equally: Jason M. Berk, Christopher Lim. ✉email: yong.xiong@yale.edu; mark.hochstrasser@yale.edu

The ubiquitin-proteasome system (UPS) comprises an essential, highly conserved set of protein modification and degradation pathways in eukaryotes[1]. Degradation of specific proteins is critical for countless cellular functions and begins with the covalent modification of substrate proteins with ubiquitin. Substrate attachment occurs between the ubiquitin carboxy-terminal glycine and a substrate residue (typically lysine). Ubiquitin itself bears seven lysine residues that can be ubiquitylated, in addition to its N-terminal α-amino group, leading to the formation of various ubiquitin polymers[1]. Two of the most common polymers are linked through the K48 or K63 residues of ubiquitin. K48 poly-ubiquitin-chain attachment frequently leads to degradation of the substrate by the proteasome, while K63 poly-ubiquitin chains are known signals for autophagy, intracellular membrane protein trafficking, and DNA repair[2–4].

To regulate ubiquitin signaling and to recycle ubiquitin, an array of deubiquitylases (DUBs) disassembles ubiquitin chains and cleaves ubiquitin from target proteins[5]. Eukaryotes have DUBs that fall into at least seven different sequence classes[6–8]. Interestingly, many bacteria that infect eukaryotic cells encode DUBs that are secreted into the host cytoplasm[9]. For pathogenic bacteria to successfully enter and thrive in host cells, they have evolved mechanisms to manipulate the host cell by injecting effector proteins into the host through specialized secretion systems[10].

Numerous bacterial effectors alter host–ubiquitin conjugates and allow the resident bacteria to obtain nutrients, suppress the innate immune response, or prevent their autophagic destruction[9]. The effectors themselves can be UPS factors, which have either an E3 ubiquitin ligase or DUB domain. The CE-clan/Ulp1-like proteases constitute the main family of characterized bacterial DUB effectors[11–16]. Eukaryotic Ulp1-like proteases instead exhibit activity toward the ubiquitin-like proteins (Ubls) SUMO or NEDD8[17,18].

In contrast to their eukaryotic counterparts, Ulp1-like bacterial effectors show distinct activities derived from the same cysteine nucleophile-based catalytic core. Most have been described with cleavage specificity for ubiquitin (for ubiquitin chains, the preference is almost always for K63-linked ones) or NEDD8, and several have acetyltransferase activity[11]. The CE-clan/Ulp1-like effector ChlaDUB1 from *Chlamydia* can both cleave ubiquitin–Ubl conjugates and attach acetyl groups to lysines of target proteins[11,19]. Given that bacteria do not themselves possess a functional UPS, bioinformatic identification of UPS enzyme domains is a useful method for finding potential effectors.

Recent analyses of DUB domain-containing proteins from obligate intracellular bacteria[11,16] motivated our in silico searches for additional candidates. Several proteins with putative Ulp1-like/CE-clan protease domains across the *Anaplasmataceae* and *Rickettsiaceae* families of intracellular α-proteobacteria were identified. Here, we succeed in determining the crystal structure of the DUB domain from *Orientia tsutsugamushi* OTT_1962 (WP_012462337.1). Very few studies have been done on the effector proteins of this pathogen[20–25].

*Orientia tsutsugamushi* causes scrub typhus, a febrile tropical disease endemic to Southeast Asia with roughly one million new cases annually. This neglected disease is acquired through transmission of the bacteria from infected *Leptotrombidium* mites. Symptoms range from asymptomatic to organ failure and death[26]. Reported cases are spreading worldwide[27], and current antibiotics are not always effective[28]. With a new potential vector[29] and a new pathogenic species (*O. chuto*) recently identified[30], it is critical that we better understand the molecular mechanisms of *Orientia* infection.

Here we report biochemical and structural data on the DUB domain of *O. tsutsugamushi* OTT_1962, hereafter called OtDUB.

Besides the predicted structure of the Ulp1-like domain, we characterize a unique ubiquitin-binding domain (UBD) in OtDUB with highly unusual properties. The UBD alters the substrate preferences of the DUB domain, and provides one of three closely positioned ubiquitin-binding sites in OtDUB. Notably, ubiquitin binding induces a transition in the UBD from a poorly folded to well-ordered state; despite this entropic cost, the UBD has an exceptionally high affinity for mono-ubiquitin. DUB and UBD activities are conserved in the related pathogen *O. chuto*. Our data reveal an unusual mechanism for ubiquitin-chain selectivity by DUBs and a remarkably high ubiquitin-binding affinity for a UBD, which can be exploited for protein ubiquitylation analyses. Characterization of the structure and enzymatic activity of OtDUB also makes the protein an attractive drug target in the treatment of scrub typhus.

## Results

**OtDUB has a divergent Ulp1-like domain.** The predicted Ulp1-like domain of OtDUB is near the N terminus of the protein, while the remainder of the 1369-residue protein is devoid of additional high-confidence predictions using standard search algorithms. Primary sequence analysis revealed conservation of a DUB-like catalytic triad (His76, Asp96, Cys135) and Trp77, suggesting that the Ulp1-like domain is catalytically active and not an acetyltransferase, respectively[11]. We cloned the N-terminal domain of OtDUB from genomic DNA of the *O. tsutsugamushi* Ikeda isolate and included residues past the putative DUB domain (1–311) to examine a potential accessory domain that could modulate DUB activity[11,18].

We determined the crystal structure of the apo-enzyme at 2.0 Å resolution, which revealed that the Ulp1-like domain of OtDUB has the predicted core fold of cysteine endopeptidase (CE)-clan proteases (Fig. 1a, b). Within this group of proteases, there are typically three variable regions (VRs) and one constant region (CR) that together account for the S1 substrate-binding interface, which contacts the distal ubiquitin[11]. (In a di-ubiquitin unit, the proximal ubiquitin contributes the lysine to the ubiquitin-ubiquitin linkage, while the distal ubiquitin provides the C-terminal carboxylate group of Gly76.) OtDUB lacks an N-terminal VR-1; instead, the C-terminal accessory domain (residues 170–259) protrudes into the VR-1 position via an extended α-helical arm positioned close to the catalytic site, suggesting that it assists in substrate binding (Fig. 1a, c). The C-terminal region of the protein fragment, residues 260–311, was apparently disordered and not observed in the structure.

When compared to the phylogenetically closest structure, RickCE from *Rickettsia bellii*, we observed structural conservation of only one VR: the small loop in VR-3 (Fig. 1c)[11]. Besides lacking the N-terminal VR-1 present in RickCE, VR-2 of OtDUB comprises a loop and two short α-helices, while RickCE VR-2 is a large helical arm largely absent in OtDUB. These large structural differences between the two DUBs, and other prokaryotic CE-clan structures, are obvious when aligned, yet the core protease fold is conserved ($C_\alpha$ root-mean-square deviation (RMSD) 1.2–3.9 Å) (Fig. 1b). The divergence of the VRs suggests that the specificity of OtDUB differs from RickCE, which cleaves K11 and K63 di-ubiquitin chains and exhibits weak activity towards NEDD8 and K29-linked di-ubiquitin[11].

**OtDUB specifically targets multiple ubiquitin-chain linkages.** To establish if the Ulp1-like domain cleaves ubiquitin and/or any Ubls, we utilized a cleavage assay where the C terminus of each protein was modified with 7-amido-4-methylcoumarin (AMC). Release of the AMC group strongly stimulates fluorescence. $OtDUB_{1–259}$ cleaved Ub-AMC far faster than any other tested Ubl

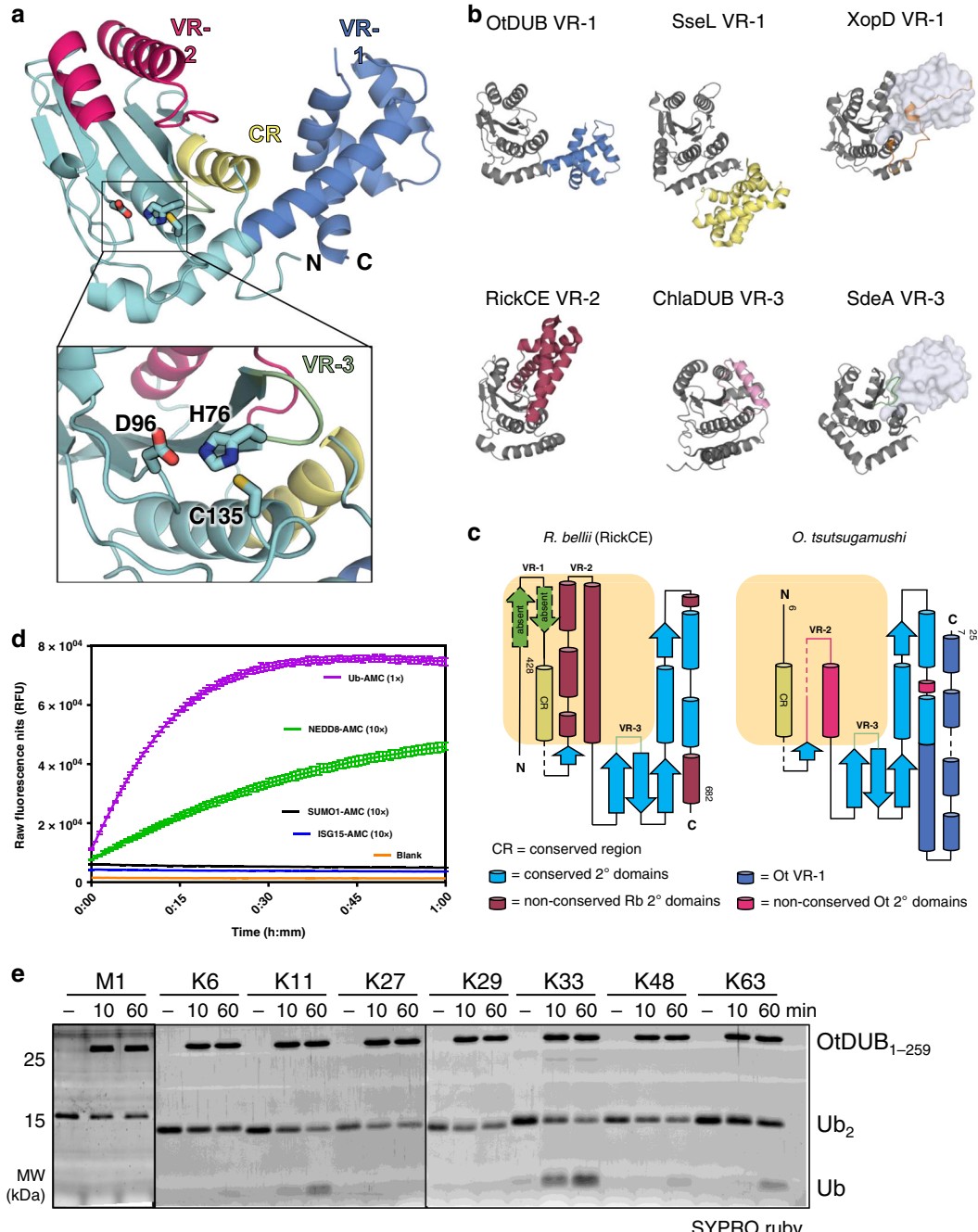

**Fig. 1 The OTT_1962 (OtDUB) Ulp1-like domain is a deubiquitylase. a** Crystal structure of OtDUB$_{1-259}$, with residues 6–257 modeled. The deubiquitylase (DUB) domain is in cyan, the proposed variable region 1 (VR-1) in slate blue, conserved region (CR) in yellow, VR-2 in magenta, and VR-3 in green (inset: Cys protease catalytic triad). **b** Structural comparison of variable regions among bacterial CE-clan DUBs (conserved catalytic fold in gray): OtDUB$_{1-259}$ VR-1 (slate blue), SseL VR-1 (yellow, PDB ID: 5HAF), XopD VR-1 (orange, PDB ID: 5JP3), RickCE VR-2 (rose, PDB ID: 5HAM), ChlaDUB1 VR-3 (violet, PDB ID: 5HAG), and SdeA VR-3 (green, PDB ID: 5CRB). S1-bound ubiquitin is shown as transparent surface where applicable. **c** Secondary structure maps of OtDUB$_{1-259}$ and the closely related DUB domain from *R. bellii*. α-Helices are represented as cylinders, and β-sheets represented as arrows. Boxed region (orange) denotes previously characterized CE-clan VRs. Structurally conserved region (CR; yellow) and VR-3 (green) are denoted. **d** Cleavage assays of Ub- and Ubl-AMC (400 nM) incubated with OtDUB$_{1-259}$ at 350 pM for Ub-AMC (1×) or 3.5 nM for Ubl-AMC (10×). Lines represent the mean of a technical triplicate with SD error bars at each 40-s time interval. **e** Representative gels for an in vitro cleavage assay using di-ubiquitin substrates (1 μM each) that were incubated in the presence or absence of OtDUB$_{1-259}$ (0.5 μM) for the indicated times, resolved by SDS-PAGE, and stained with SYPRO Ruby ($n = 2$). Black lines denote separate gels. Source data are provided as a Source Data file.

(Fig. 1d). It showed no activity toward the Ubls ISG15-AMC or SUMO1-AMC and weak activity toward NEDD8-AMC, similar to other bacterial CE-clan DUBs[11,15]. To determine if OtDUB$_{1-259}$ had any preferences for specific ubiquitin-ubiquitin-chain linkages, we screened a di-ubiquitin panel, including all seven lysine linkages and linear di-ubiquitin. OtDUB$_{1-259}$ could cleave multiple chain linkages, including K6, K11, K33, K48, and K63 (Fig. 1e). A preference was observed for K33-linked chains, a poorly studied linkage type associated with immune responses and secretory pathway trafficking[31–33]. Given the overwhelming

roles of K48- and K63-linked chains in pathogen defense and known modulation of these chain types by bacterial effectors[34,35], we pursued further analysis of chains with these linkages.

We suspected that the DUB might cleave longer ubiquitin chains more efficiently and tested K63 and K48 tri- and tetra-ubiquitin as substrates. Indeed, various OtDUB fragments (1–177, 1–259, 1–311) cleaved extended K48 and K63 chains significantly faster than di-ubiquitin (Supplementary Figs. 1a, b and 2a vs. c). Therefore, OtDUB is an active DUB with preference for ubiquitin chains longer than dimers.

**Crystal structure of OtDUB in complex with ubiquitin.** Our biochemical assays revealed a UBD within the noncanonical VR-1 of OtDUB (Supplementary Note 1, Supplementary Fig. 1b, c). To analyze the structural basis of the substrate preferences of OtDUB, we assembled a complex of OtDUB$_{1-259}$ with free ubiquitin and determined the crystal structure of the complex at 2.2 Å resolution. Three ubiquitin molecules bound to each OtDUB fragment (Fig. 2a and Supplementary Fig. 3). One ubiquitin is bound to the active site (Ub$_{S1}$, gray) and represents the distal ubiquitin that would remain after cleavage of the proximal ubiquitin at the S1′ site (absent from the structure, Fig. 2b). The position of Ub$_{S1}$ in the OtDUB complex is rotated nearly 90° relative to the position of Ub$_{S1}$ in complexes with XopD and SdeA (Fig. 2c). Overall, Ub$_{S1}$ binding to OtDUB buries an extensive interface of nearly 1200 Å$^2$, and results in modest conformational changes in OtDUB (overall RMSD < 1.5 Å) attributable mostly to a slight (~10°) rotation of VR-1 towards Ub$_{S1}$. We also observed the density for two additional ubiquitin molecules: one bound to the side of the OtDUB that we propose represents the S2 site (pink, Fig. 2), and a third ubiquitin molecule (yellow) bound to the backside of VR-1 far from the active site, which likely represents the biochemically identified UBD (Fig. 2a, Supplementary Fig. 1c).

**S2 and S1 ubiquitins are positioned for K63 linkage.** The discovery of an additional ubiquitin adjacent to the active site Ub$_{S1}$ motivated us to examine the role of the second binding site in enzyme activity and substrate recognition. The interface between Ub$_{S1}$ and the second ubiquitin (pink) bound to the side of OtDUB places the C terminus (R74, red sphere) of the latter ubiquitin near K63 of Ub$_{S1}$ (Figs. 2b and 3a). We hypothesized that this second binding site represents an S2 site involved in binding the distal-most ubiquitin of a K63-linked ubiquitin tri-mer[6]. The OtDUB:Ub$_{S2}$ interface is composed of a hydrophobic pocket (OtDUB L55/F59/I90/C116) centered around L8 in the β-hairpin loop of Ub$_{S2}$ and electrostatic interactions between OtDUB D57/D88 and R42/K6 of ubiquitin around the periphery.

To test whether the S2-bound ubiquitin mimics distal ubiquitin binding in K63-linked poly-ubiquitin, we performed cleavage assays with K63- and K48-linked Ub$_4$ chains with either wild-type (WT) OtDUB$_{1-259}$ or a structure-guided point mutant (F59T) designed to weaken the S2 site interface (Fig. 3b). WT enzyme cleaved K63- and K48-linked poly-ubiquitin with similar efficiencies as measured by loss of tetra-ubiquitin (Fig. 3c, d). By contrast, the F59T mutation significantly decreased the percentage of K63-linked tetra-ubiquitin cleaved, but caused only small decreases in activity against K48-linked poly-ubiquitin at early time points. These data bolster the assertion that the ubiquitin bound to the side of VR-2 (the S2 site) represents the most distally bound ubiquitin in a K63-linked chain ($n \geq 3$) and provide a structural explanation for how extended K63-linked ubiquitin chains are accommodated in the OtDUB structure. K48 chains probably can also bind VR-2, but this would require that

the Ub$_{S2}$ be rotated and make different surface contacts with the DUB.

When the F59T S2 site mutant was analyzed for Ub-AMC cleavage, it exhibited WT cleavage kinetics (Fig. 3e); K48 di-ubiquitin cleavage kinetics were also unaffected (Fig. 3f, g). Interestingly, K63 di-ubiquitin cleavage was accelerated by OtDUB$_{1-259}$-F59T compared to WT, likely because the S2 site can no longer bind efficiently to the distal ubiquitin, which would otherwise compete with the S1 site for binding this ubiquitin in K63 di-ubiquitin. Our data imply that the identified S2 site binds K63 chains preferentially and assists in accelerating K63 long-chain cleavages while inhibiting K63 di-ubiquitin cleavage.

**Noncanonical VR-1 modulates binding of ubiquitin-chain types.** We next examined the S1 site to determine if any of the interactions with Ub$_{S1}$ could influence substrate specificity. The C terminus of Ub$_{S1}$ adopts an extended conformation and is positioned by a highly conserved di-acidic motif (OtDUB E16/D17), present in the CR of all CE-clan proteases, and by hydrophobic residues (OtDUB H38/H71/W77) that recognize L73 of ubiquitin (Fig. 4a, left panel). The Ub$_{S1}$ itself is sandwiched between two OtDUB regions: on one side it contacts VR-2, which creates a hydrophobic pocket (OtDUB I36/V39/L46/T50) for G47 of ubiquitin, and on the other, VR-1 uses charged residues (OtDUB R196/E238/R242) to interact with oppositely charged residues of ubiquitin, a noncanonical ubiquitin surface for S1 site interactions (Fig. 4a, right panel). In contrast to its structural compatibility with binding K63 di-ubiquitin chains (Fig. 2b), when Ub$_{S1}$ is aligned with the proximal ubiquitin in K48 di-ubiquitin from published K48-chain structures, the distal ubiquitin clashes with the DUB domain (Supplementary Fig. 4a). Nevertheless, OtDUB cleaves K48-linked ubiquitin chains (Fig. 3c, f), suggesting that it repositions K48-linked chains for catalysis using distinct determinants.

Since OtDUB VR-2 only interacts with G47 in ubiquitin and does not specifically contact K48, we hypothesized that VR-1 positions Ub$_{S1}$ in a way that is compatible with both K63- and K48-linked chains. To test if VR-1 affects cleavage specificity, we mutated its ubiquitin-interacting residues (R196A/E238A and E238A/R242A double mutants; Fig. 4a) and analyzed the mutants in cleavage assays with K48- and K63-linked tetra-ubiquitin (Fig. 4b). Both OtDUB double mutants inhibited K48-chain cleavage, showing a significant impairment by 10 min (Fig. 4c). By contrast, cleavage of K63-linked chains was not affected. Analysis of these mutations with Ub-AMC also revealed a slight decrease in enzyme activity (Fig. 4d), but the E238A/R242A mutant had no significant effect on K63 or K48 di-ubiquitin cleavage (Fig. 4e, f). These data show that the noncanonical VR-1 is required for full DUB activity against extended chains and suggest that ubiquitin binding at the K63-chain-favoring S2 site can override impaired binding to the S1 site by long K63 chains but not K48 chains.

**OtDUB has a high-affinity UBD.** The crystal structure revealed a third, independent ubiquitin molecule bound to the backside of VR-1, which we show is a high-affinity ubiquitin-binding inter-face. VR-1 folds into a compact helical bundle that presents two antiparallel helices for interaction with the canonical I44 surface patch of ubiquitin, the most common site for UBD interactions (Fig. 2a)[36]. This backside VR-1 or UBD site binds ubiquitin using a central hydrophobic groove (V203/F207/L221/L225) that interacts with all three highly conserved residues on the I44 patch of ubiquitin (L8/I44/V70) (Fig. 5a), and the hydrophobic core is encircled by negatively charged and polar residues (OtDUB

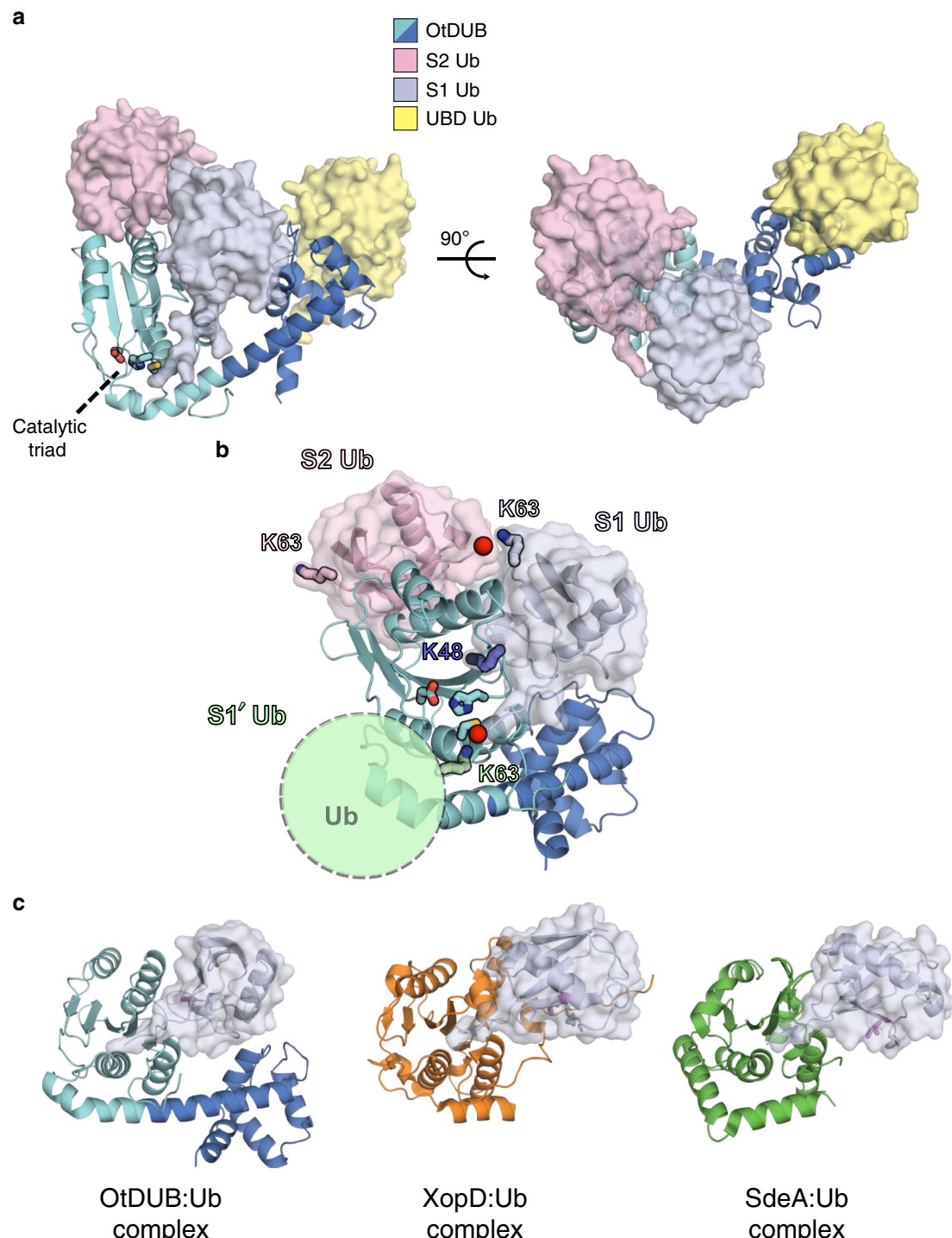

**Fig. 2 OtDUB$_{1-259}$–ubiquitin crystal complex reveals three distinct ubiquitin-binding sites. a** Overall structure of the OtDUB$_{1-259}$–ubiquitin complex with OtDUB$_{1-259}$ in cartoon representation and individual ubiquitin molecules in cartoon/surface representation. Catalytic triad residues shown as sticks in **a**, **b**. **b** Schematic of the potential K63 ubiquitin-chain linkage, highlighting the arrangement of two neighboring ubiquitin molecules bound to OtDUB$_{1-259}$. K63 side chains shown as sticks and C-terminal glycines shown as red spheres. **c** Structural comparison of OtDUB$_{1-259}$–S1 ubiquitin complex with other bacterial DUB–ubiquitin complexes. For XopD-Ub (orange, PDB ID: 5JP3) and SdeA–Ub (green, PDB ID: 5CRB), DUBs are shown as cartoon, S1 ubiquitin is shown as cartoon/surface, and Ile44 is shown as magenta sticks.

D204/D208/N222/D226) that interact with positively charged residues in ubiquitin (Fig. 5c).

To test whether these residues were important for binding ubiquitin in solution, we employed glutathione *S*-transferase (GST)-pulldown assays with purified proteins bearing mutations in either the GST-tagged UBD$_{170-259}$ or ubiquitin. Mutation of

hydrophobic residues in the OtDUB$_{UBD}$ abolished (V203D and L221D) or weakened (F207D) binding (Fig. 5b, left). Alanine mutation of any one of the three hydrophobic residues in the ubiquitin I44 patch reduces or abolishes binding to most UBDs[37–41]. Strikingly, none of these single ubiquitin mutations reduced OtDUB$_{UBD}$ binding (Fig. 5b). When we examined the

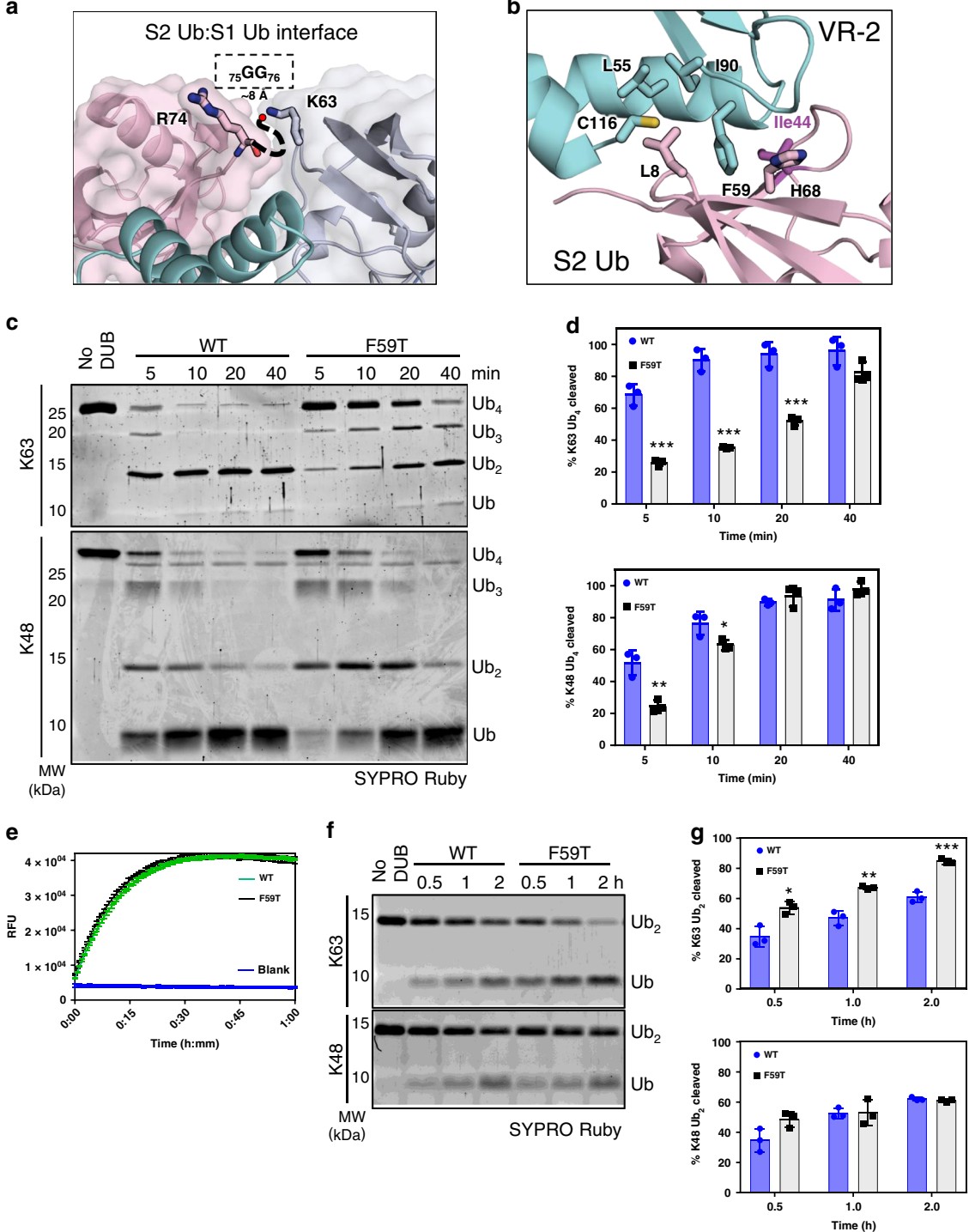

**Fig. 3 The S2 ubiquitin-binding site controls preference for K63-linked chain cleavage. a** Positioning of the C terminus of S2 ubiquitin (pink) and proximity to K63 of the central S1 ubiquitin (light gray) suggests a preferred orientation for K63 chains. Terminal glycines are missing from the density due to inherent flexibility. **b** Detailed view of interactions at the S2 site. OtDUB VR-2 is in cyan, and S2 ubiquitin is in pink. **c** In vitro cleavage assays of K63 and K48 tetra-ubiquitin (Ub$_4$) chains (2 μM) by WT and F59T OtDUB$_{1-259}$ (50 nM). Source data are provided as a Source Data file. **d** Quantification of Ub$_4$ cleavage rates using data, exemplified by **c**, from three independent experiments. Mean values and SD (error bars) are shown. Source data are provided as a Source Data file. **e** Comparison of Ub-AMC cleavage by WT and F59T OtDUB$_{1-259}$. Ub-AMC (400 nM) was incubated alone (Blank) or with 350 pM of the indicated OtDUB. Lines are the average of technical triplicates with SD bars at each 40-s interval. **f** K63 and K48 di-ubiquitin cleavage by WT and F59T OtDUB$_{1-259}$. Di-ubiquitin (1 μM) was incubated in the presence or absence of the indicated OtDUB (0.5 μM) for 0.5, 1, and 2 h. Source data are provided as a Source Data file. **g** Quantitative comparison of di-ubiquitin cleavage rates using data, exemplified by **f**, from three experiments; mean and SD values are shown. Source data are provided as a Source Data file. Unpaired, two-tailed $t$ tests were performed (**d**, **g**) for comparisons between OtDUB$_{1-259}$ WT and F59T for each condition and time point (*$p < 0.05$, **$p < 0.005$, ***$p < 0.0005$).

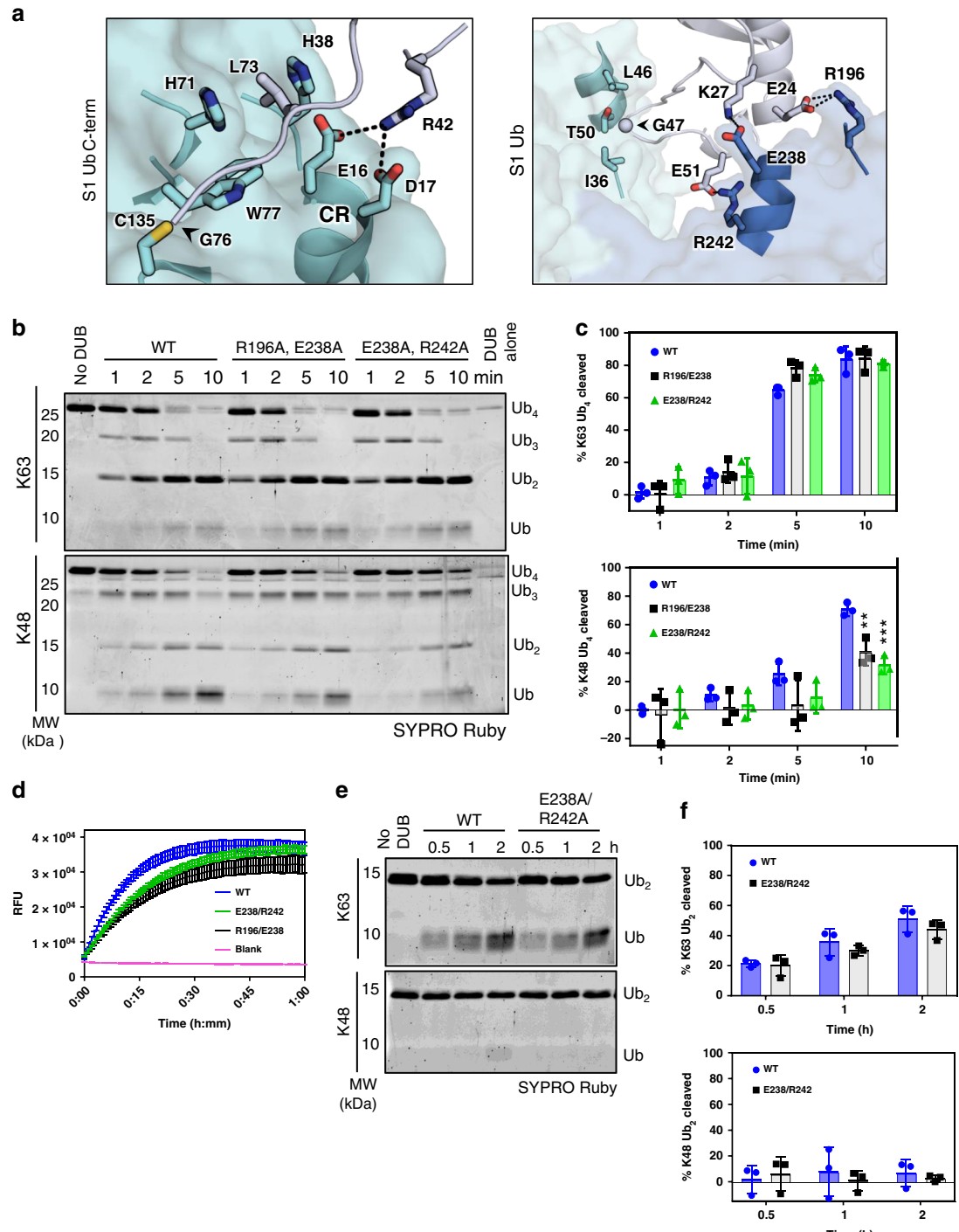

**Fig. 4 Interactions with OtDUB VR-1 are required for efficient K48-chain cleavage. a** Two views of S1 ubiquitin (gray) in the active site of OtDUB (cyan). Left: C terminus of S1 ubiquitin in an elongated conformation with interacting residues shown as sticks. Right: VR-2 (cyan) and VR-1 (slate) sandwich S1 ubiquitin using hydrophobic and electrostatic interactions, respectively. **b** In vitro cleavage assays of tetra-ubiquitin chains (2 μM) by WT and mutants of OtDUB$_{1-259}$ (50 nM). Source data are provided as a Source Data file. **c** Quantification of Ub$_4$ cleavage rates using data, exemplified by **b**, from three independent experiments; mean and SD are shown. Source data are provided as a Source Data file. Unpaired, two-tailed $t$ tests were performed for comparisons between OtDUB$_{1-259}$ WT and VR-1 mutants for each condition and time point (**$p < 0.005$, ***$p < 0.0005$). **d** Ub-AMC-cleavage assays comparing the activities of WT, R196A/E238A and E238A/R242A OtDUB$_{1-259}$. Ub-AMC (400 nM) was incubated alone (Blank) or with 350 pM of the indicated OtDUB. Lines are the average of technical triplicates with SD bars at each 40-s interval. **e** K63 and K48 di-ubiquitin cleavage assays utilizing WT and E238A/R242A OtDUB$_{1-259}$. Di-ubiquitin (1 μM) was incubated in the presence or absence of the indicated OtDUB (0.5 μM) for 0.5, 1, and 2 h. Source data are provided as a Source Data file. **f** Quantification of Ub$_2$ cleavage rates using data, exemplified by **e**, from three independent experiments; mean with SD values are shown. Source data are provided as a Source Data file.

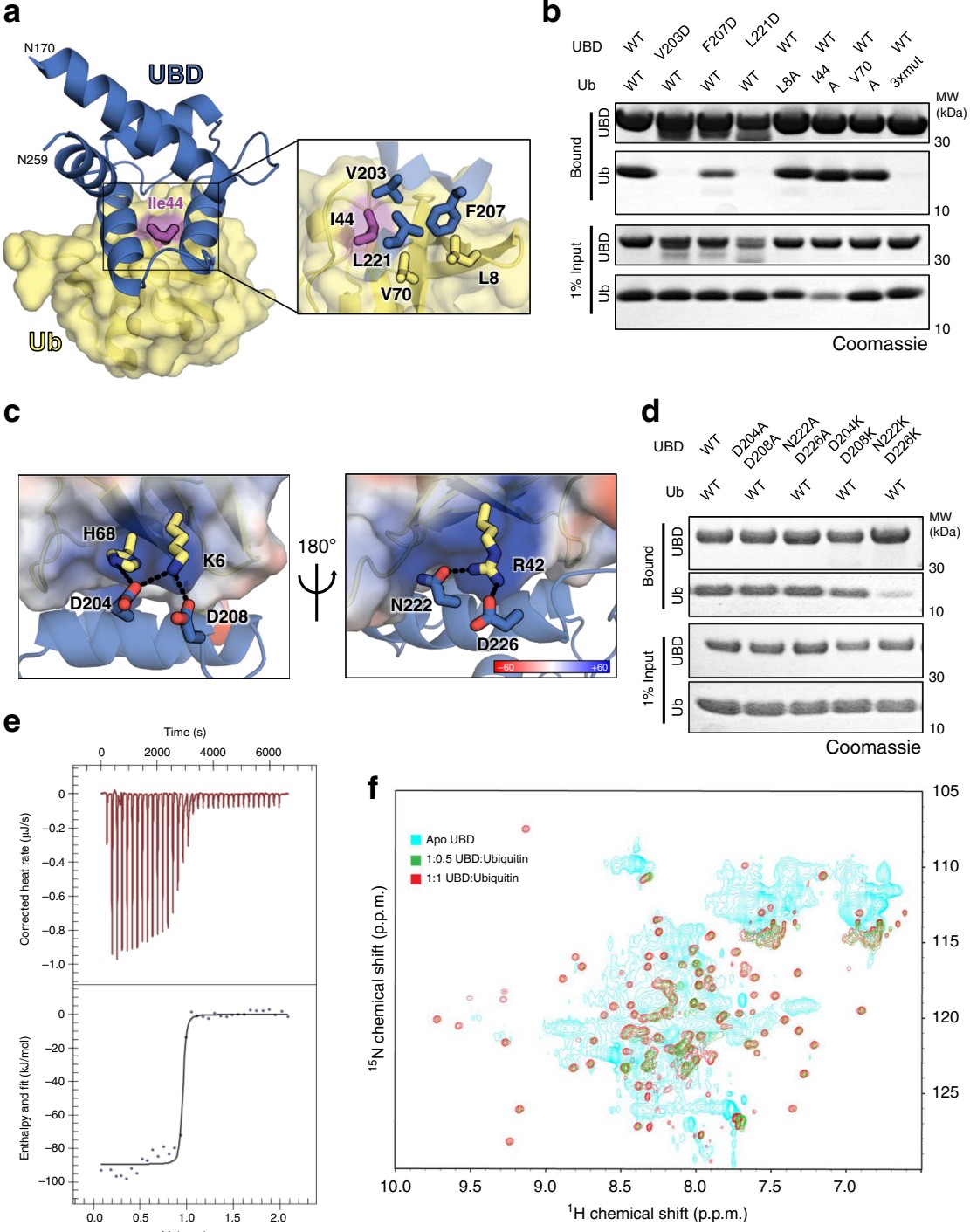

**Fig. 5 The OtDUB_UBD has unusually high affinity for ubiquitin. a** Detailed view of hydrophobic interactions between the OtDUB_UBD (blue) and ubiquitin (yellow). **b** Glutathione S-transferase-pulldown experiments between GST-UBD_{170–259} (25 μM) and 6His-ubiquitin (75 μM) or mutants thereof. Protein mixtures were combined and incubated with glutathione-charged resin, washed extensively, and bound complexes were eluted and resolved by SDS-PAGE (n = 2). Source data are provided as a Source Data file. **c** Detailed views of electrostatic interactions between the OtDUB_UBD (blue cartoon and sticks) and ubiquitin (shown in transparent electrostatic potential surface). Positive and negative charges are in blue and red, respectively. **d** As in **b**, but with dual charge-neutralizing or charge-reversed mutants of the ubiquitin-binding interface of the UBD (n = 2). Source data are provided as a Source Data file. **e** Representative ITC titration of UBD with ubiquitin. Upper panel shows raw injection data over time, and the lower panel shows integrated heats over the course of the reaction (n = 3). Performed with 140 μM mono-ubiquitin being titrated into 20 μM of the OtUBD_{170–264}. **f** $^{1}$H -$^{15}$N HSQC spectral overlay for the OtUBD_{170–264} domain of OtDUB: apo (pink), in 1:0.5 (turquoise), and 1:1 (dark blue) complexes with unlabeled ubiquitin. Data were obtained at 293 K, pH = 7.5, at 600 MHz. Concentrations of OtUBD_{170–264} in the samples are 1.55 mM (apo), 1.41 mM (1:0.5 complex), and 1.29 mM (1:1 complex).

electrostatic interactions at this interface, we also found evidence for strong interactions. Charge-neutralizing mutations (D204A/D208A and N222A/D226A) did not reduce binding to ubiquitin, and even charge reversals could not fully disrupt binding (Fig. 5d). These analyses suggest that the OtDUB$_{UBD}$ binds ubiquitin with high affinity.

We used isothermal titration calorimetry (ITC) to quantify the ubiquitin–OtDUB$_{UBD}$ interaction (Fig. 5e). Titration of OtDUB$_{170-264}$ (the UBD) with ubiquitin revealed a dissociation constant ($K_d$) of $5 \pm 4$ nM and a stoichiometry of $n = 0.9 \pm 0.1$ (Table 1). The $K_d$ of OtDUB$_{UBD}$ for mono-ubiquitin is more than two orders of magnitude tighter than for any previously reported natural UBD[42,43]. While many UBDs preferentially bind specific ubiquitin linkages[44], based on the stoichiometry and our structural information, this is likely not the case for OtDUB$_{UBD}$. Our data indicate the UBD binds mono-ubiquitin and likely does not prefer particular ubiquitin-chain types.

We were unable to obtain crystals of the OtDUB$_{UBD}$ by itself or in complex with ubiquitin for structural analysis. We turned instead to nuclear magnetic resonance (NMR). Unexpectedly, 2D-NMR analysis of the UBD$_{170-264}$ alone revealed a broad and ill-defined backbone spectrum, suggestive of conformation heterogeneity and disorder (Fig. 5f). When ubiquitin was titrated in at equimolar amounts, the spectrum originally characterized by broadened, low-intensity resonances shifted to a dispersed and well-resolved spectrum containing many new resonance peaks, characteristic of a fully folded structure. A substantial structural transition of the UBD was supported by the apo-DUB-UBD$_{1-311}$ crystal structure (Fig. 1a), which lacked density for residues 224–235 at the beginning of the UBD, a region that includes the ubiquitin-interacting residues D226 and K230. These data are also consistent with the large, negative $\Delta S$ calculated from ITC measurements for the binding reaction between UBD and ubiquitin (Table 1), indicating a large reduction in entropy upon binding. This entropic penalty to binding is overcome by a very large reduction in enthalpy.

The ubiquitin residues that bind to OtDUB$_{UBD}$ are identical at seven of eight corresponding positions in NEDD8; we therefore tested binding of the UBD to this Ubl (Supplementary Fig. 5a). By size-exclusion chromatography (SEC), we observed a complete shift of NEDD8 to an earlier elution position, indicative of tight binding (Supplementary Fig. 5b). By ITC, the UBD–NEDD8 interaction had a $K_d$ of $31 \pm 14$ nM (Table 1, Supplementary Fig. 5c). The physiological relevance of this interaction is uncertain inasmuch as cellular concentrations of ubiquitin are much higher than those of NEDD8 and the OtDUB$_{UBD}$:ubiquitin association is tighter. Only a handful of UBDs have been shown to bind multiple Ubls[45], so we determined if the OtDUB$_{UBD}$ was an indiscriminate Ubl/ubiquitin binder by looking at its interaction with the Ubl SUMO2. Human SUMO2 did not bind the UBD or the DUB domain of OtDUB based on SEC analysis with OtDUB$_{1-311}$ (Supplementary Fig. 5d)[45]. Together, these results demonstrate that the OtDUB$_{UBD}$ binds specifically to both mono-ubiquitin and NEDD8 with high affinity and exhibits selective Ubl binding.

**Ubiquitin binding by the UBD alters OtDUB activity.** Because ubiquitin (Ub$_{UBD}$) stabilizes the UBD fold, it should affect the contribution of the opposing VR-1 surface to ubiquitin binding at the S1 site (Fig. 4). We therefore tested for effects on OtDUB DUB activity. K48 di-ubiquitin was cleaved faster by the OtDUB$_{1-259}$-V203D mutant, a variant that abolishes UBD-Ub$_{UBD}$ binding (Fig. 5b) and was indifferent to ubiquitin pre-incubation (Fig. 6a, b). When WT OtDUB$_{1-259}$ was preincubated with ubiquitin, it cleaved K48 di-ubiquitin faster than the V203D

mutant (Fig. 6a, b). These results suggest the UBD sequesters K48 di-ubiquitin away from the active site, an effect that is blocked by pre-binding the UBD with ubiquitin. The higher activity of WT OtDUB$_{1-259}$ toward K48 di-ubiquitin when preincubated with ubiquitin is likely due to VR-1 fully folding when the UBD is occupied (Fig. 5f), while only partially folding in the context of V203D. To test directly the contribution of ubiquitin binding to the S1 site surface from VR-1, WT OtDUB$_{1-259}$ and the E238A/R242A derivative were incubated with ubiquitin prior to K48 di-ubiquitin cleavage. As expected, the E238A/R242A mutant showed significantly reduced cleavage for K48 (Fig. 6c, d), and had no effect on K63 di-ubiquitin cleavage (Fig. 6c, Supplementary Fig. 6b).

In contrast to the enhanced K48 di-ubiquitin cleavage by OtDUB$_{1-259}$-V203D, K63 di-ubiquitin was cleaved significantly more slowly (Fig. 6a, Supplementary Fig. 6a). The exact reason for this was unclear. When the V203D variant was assayed for Ub-AMC cleavage, we also see a reduced activity compared to OtDUB$_{1-259}$; the activity was identical to that of OtDUB$_{1-177}$, which lacks the UBD altogether (see Fig. 6f). This reduced activity between OtDUB$_{1-177}$ and OtDUB$_{1-259}$ was also observed for di-ubiquitin cleavage (Supplementary Fig. 2c). Together, these results suggest that a fully intact S1 site is required for optimal K63 di-ubiquitin cleavage and Ub-AMC cleavage, although the mechanistic details remain to be fully explained.

The lowered activity against K63 di-ubiquitin, but not K48 chains, when OtDUB was preincubated with ubiquitin might be due at least in part to the preference of the S2 site for binding K63 chains, which would compete for binding the distal ubiquitin of K63 di-ubiquitin and prevent the isopeptide linkage from reaching the active site. To test this, we analyzed the ability of the S2 site mutant OtDUB$_{1-259}$-F59T to cleave K63 di-ubiquitin. As predicted, the F59T mutant cleaved K63 di-ubiquitin significantly faster than WT (Fig. 6e, g). Preincubation with ubiquitin further accelerated K63 di-ubiquitin cleavage by F59T, which could be explained by either a fully intact S1 site or reduced substrate competition with the UBD. No differences were seen with K48 di-ubiquitin cleavage by F59T (Fig. 6e, Supplementary Fig. 6c).

We also tested K63 and K48 tetra-ubiquitin-chain cleavage by the V203D mutant. The mutant and WT OtDUB$_{1-259}$ (at 50 nM) showed similar cleavage kinetics for the K48 chains (Supplementary Fig. 6d), but the V203D mutant had a weak but statistically significant delay in K63 cleavage (Supplementary Fig. 6e), in line with the K63 di-ubiquitin and Ub-AMC results.

By contrast, the V203D mutant at high enzyme concentration (500 nM) enhanced cleavage kinetics to a degree equivalent to OtDUB$_{1-177}$, which lacks the UBD, and this mutant did not display the self-inhibition that can be alleviated by preincubation with ubiquitin (Fig. 6h [K63], Supplementary Fig. 6f [K48]). Thus, at high enzyme concentration, OtDUB$_{1-259}$-V203D and OtDUB$_{1-177}$ cleave both chain types more efficiently than WT OtDUB$_{1-259}$, which is negatively regulated by the UBD.

**Table 1 Thermodynamic parameters for OtDUB$_{UBD}$ binding Ub and NEDD8 as determined by ITC.**

|  | **Mono Ub** | **NEDD8** |
|---|---|---|
| $K_d$ (nM) | $5.3 \pm 3.9$ | $31 \pm 14$ |
| $n$ | $0.9 \pm 0.1$ | $0.9 \pm 0.1$ |
| $\Delta H$ (kJ/mol) | $-90.2 \pm 4.6$ | $-61.0 \pm 4.1$ |
| $\Delta S$ (J/mol•K) | $-142.4 \pm 21.7$ | $-60.0 \pm 17.3$ |
| $-T\Delta S$ (kJ/mol) | $42.5 \pm 6.5$ | $17.9 \pm 5.2$ |
| $\Delta G$ (kJ/mol) | $-47.7 \pm 1.9$ | $-43.1 \pm 1.2$ |

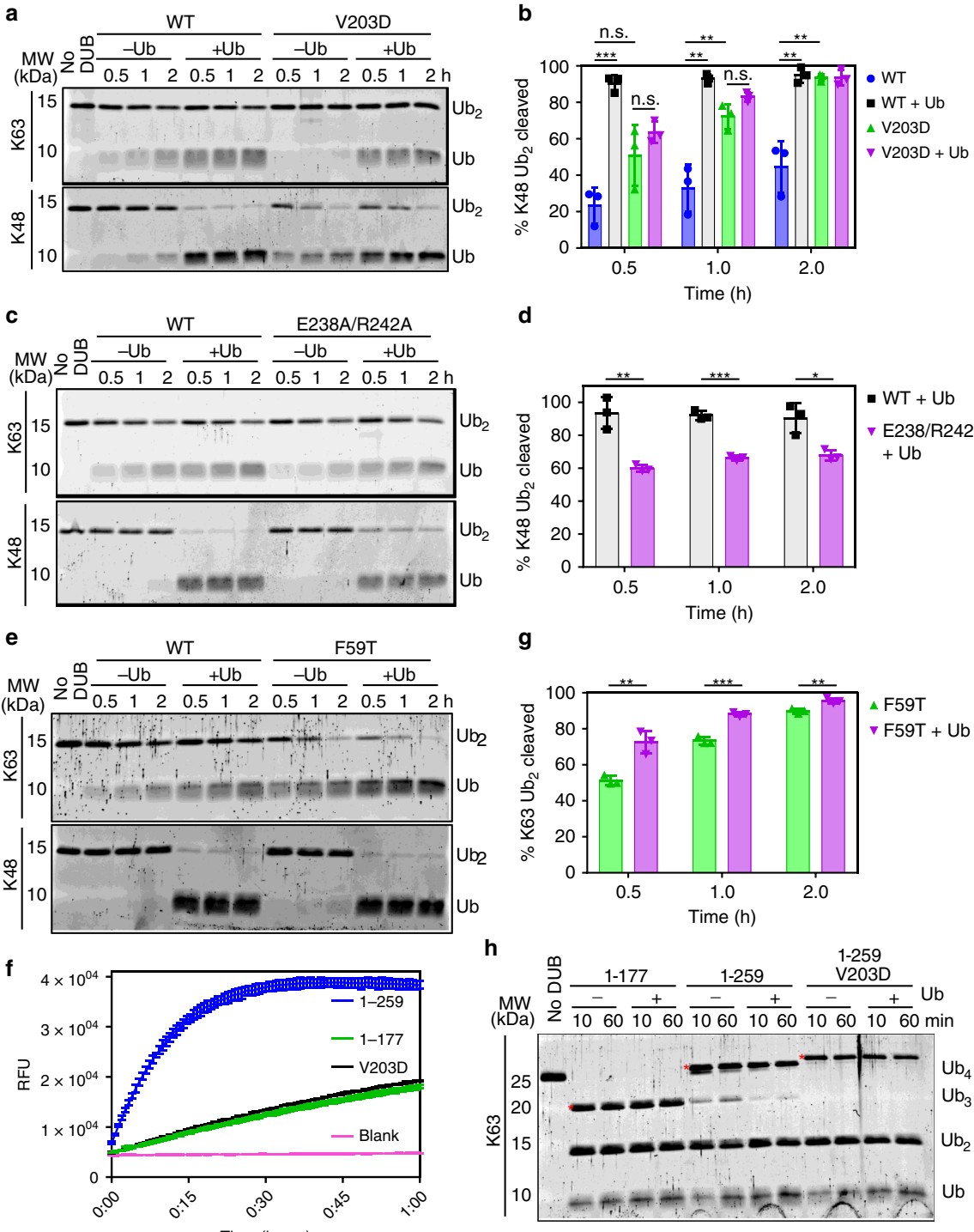

**OtDUB activities are conserved among *Orientia* species**. A second pathogenic *Orientia* species, *O. chuto*, has been isolated recently[30]. Sequence alignment of an OtDUB-homologous protein in *O. chuto* (WP_052694629.1) revealed 66% identity to the OtDUB$_{1-259}$ fragment, including complete conservation in and around the catalytic triad (Fig. 7a). We expressed and purified the *O. chuto* DUB fragments OcDUB$_{1-177}$ and OcDUB$_{1-299}$. Both OcDUB constructs exhibited strong activity toward Ub-AMC, even higher than each of the comparable OtDUB constructs (Fig. 7b). When assayed against all linkage types, OcDUB$_{1-299}$ exhibited specificity comparable to OtDUB$_{1-259}$, but again with higher activity (Fig. 7c vs. 1e). Similarly, OcDUB$_{1-299}$ cleaved

K63-linked tetra-ubiquitin at a faster rate than OtDUB (Fig. 7d). However, compared to OtDUB, the OcDUB$_{1-299}$ cleaved extended K48 chains less efficiently (build-up of tri-ubiquitin), and OcDUB$_{1-177}$ was less efficient at cleaving both K48 and K63 tetra-ubiquitin chains (Fig. 7d). Taken together, these data suggest that OcDUB and OtDUB ubiquitin-chain recognition and processing are significantly different despite their high sequence similarity.

To look further into these differences, we tested OcDUB for ubiquitin binding by the elements homologous to the OtDUB$_{UBD}$ and S2 site. The UBD$_{170-299}$ of OcDUB also bound mono-ubiquitin stably based on SEC (Fig. 7e). This is consistent with the

**Fig. 6 The UBD domain is critical for optimal cleavage of K48 and K63 di-ubiquitin substrates. a** WT or V203D OtDUB$_{1-259}$ was preincubated with an equimolar concentration of ubiquitin then diluted to 0.5 with 1 μM K63 or K48 di-ubiquitin. Samples were resolved by SDS-PAGE and SYPRO Ruby stained. Source data are provided as a Source Data file. **b** Quantification of K48 di-ubiquitin cleavage from **a** with the mean of three experiments and SD bars are shown. Source data are provided as a Source Data file. **c** WT or E238A/R242A OtDUB$_{1-259}$ was preincubated with an equimolar concentration of ubiquitin and then diluted to 0.5 with 1 μM K63 or K48 di-ubiquitin. Samples were processed as in **a**. Source data are provided as a Source Data file. **d** Quantification of K48 di-ubiquitin cleavage from **c** with the mean of three experiments and SD bars are shown. Source data are provided as a Source Data file. **e** WT or F59T OtDUB$_{1-259}$ was preincubated with an equimolar concentration of ubiquitin and then diluted to 0.5 with 1 μM K63 or K48 di-ubiquitin. Samples were processed as in **a**. Source data are provided as a Source Data file. **f** Ub-AMC-cleavage assay comparing the activities of OtDUB$_{1-177}$, WT, and V203D OtDUB$_{1-259}$. Ub-AMC (400 nM) was incubated alone (Blank) or with the indicated OtDUB (350 pM). Lines represent the average of a technical triplicate with SD bars at each 40 s time point. **g** Quantification of K63 di-ubiquitin cleavage from **e** with the mean of three experiments and SD bars are shown. Source data are provided as a Source Data file. **h** OtDUB$_{1-177}$, OtDUB$_{1-259}$, and OtDUB$_{1-259}$-V203D were preincubated with an equimolar concentration of ubiquitin before adding to 1 μM of K63 tetra-ubiquitin chains to obtain 0.5 μM of enzyme. Samples were processed as in **a** ($n = 2$). Red asterisks indicate enzyme protein bands. Source data are provided as a Source Data file. Unpaired, two-tailed $t$ tests were performed (**b**, **d**, **g**) for pair-wise comparisons between OtDUB$_{1-259}$ WT and mutant variants for each condition and time point (\*$p < 0.05$, \*\*$p < 0.005$, \*\*\*$p < 0.0005$, n.s. = not significant).

high conservation of ubiquitin-interacting residues between the *Orientia* UBDs (Fig. 7a, boxed red).

To test for an S2 site in OcDUB, we compared the ubiquitin-binding capacity of inactive OtDUB$_{1-177}$-C135A (which can bind ubiquitin at both S1 and S2) to OcDUB$_{1-177}$-C134A. The two protein fragments were incubated with increasing molar equivalents of ubiquitin. The mixtures were fractionated by SEC (Supplementary Fig. 7a), and equivalent fractions resolved by sodium dodecyl sulfate-polyacrylamide gel (SDS-PAGE) (Fig. 7f). OtDUB$_{1-177}$-C135A could bind two ubiquitins based on the progressive shift of the DUB protein from the D column fractions to the earlier (larger protein-containing) A and B fractions as ubiquitin levels were increased. OcDUB$_{1-177}$-C134A, on the other hand, maintained a comparable elution pattern across the titration from 1.0 to 2.5 ubiquitin equivalents and never migrated to fraction A, suggesting the absence of a second (S2) site. When the S2 site mutation F59T is introduced (OtDUB$_{1-177}$-F59T, C135A) the elution pattern mirrors OcDUB$_{1-177}$-C134A, suggesting that F59T alone is sufficient to disrupt the S2 site of OtDUB (Fig. 7f). Lack of an S2 site alone does not explain the differences between OtDUB and OcDUB, especially the efficient cleavage of K63 tetra-ubiquitin down to mono-ubiquitin. Additional work will be required to determine the basis of the different enzymatic properties of these related proteins.

Together, these results demonstrate that although the two *Orientia* DUB enzymes have diverged mechanistically, the central activities of the DUB are conserved, consistent with the protein playing an important role in infections by both species.

## Discussion

Although bacteria lack ubiquitin-conjugation systems, numerous CE-clan ubiquitin proteases are encoded by bacterial pathogens, reflecting the importance of manipulating host ubiquitin pathways for successful infection[11–16]. We have found that the Ulp1-like DUB domain from the pathogen *O. tsutsugamushi* OTT_1962 protein (OtDUB) has several unusual features. It cleaves many, but not all, lysine-linked di-ubiquitin substrates, with strong activity toward K33-linked di-ubiquitin. K33-linked ubiquitin chains have roles in T cell activation, innate immune response, and post-Golgi protein trafficking[31–33]. OtDUB could potentially disrupt any or all of these pathways during an *Orientia* infection. Our work focused on K48 and K63 chains as potentially physiological relevant modifications based on the roles they are also known to play in defenses against pathogens[34,35].

OtDUB preferentially cleaves ubiquitin chains having at least three ubiquitins and efficiently cleaves both K48- and K63-linked chains, unlike other members of this clan that prefer K63 chains. The ability to cleave both chain types can be rationalized by our structural and mutagenesis results. Binding of K63 chains at the S2 site found in VR-2 helps position such chains for cleavage.

Cleavage of both K48 and K63 chains appears to depend on contacts made with the S1 site in the noncanonical VR-1, but only K48-chain cleavage is impaired by mutating this surface; this is likely because K48 chains cannot bind optimally at the S2 site, in contrast to K63 chains. When various K48 di-ubiquitin crystal structures are aligned with OtDUB Ub$_{S1}$, there are clashes between the DUB and the distal ubiquitins (Supplementary Fig. 4a). However, the flexibility of the ubiquitin C-terminal segment allows the distal ubiquitin in a K48-type chain to adopt many conformations; indeed, two known conformations of a K48 ubiquitin dimer bound to an enzyme[46,47] can be readily modeled into the S1 site with minimal clash at VR-2 of OtDUB (Supplementary Fig. 4b). This suggests the OtDUB utilizes K48-chain flexibility in order to efficiently cleave such chains, as we observed.

A strong preference for cleaving longer ubiquitin chains has previously been seen with the MINDY-1 and ZUFSP/ZUP1 families of DUBs[7,8,48], but has not been reported for CE-clan DUBs. Both MINDY-1 and ZUFSP DUBs require multiple ubiquitin-interacting motifs for poly-ubiquitin-chain cleavage, while the OtDUB$_{UBD}$ is not required for extended K48- or K63-chain cleavage (Fig. 6h, Supplementary Figs. 2a, c and 6f).

The high-resolution structures of OtDUB reveal a previously undescribed VR architecture and extensive interactions with multiple ubiquitin molecules. Interactions of DUBs with ubiquitin have previously been analyzed structurally, typically with covalently modified ubiquitin derivatives to generate stable enzyme–substrate complexes. Our approach of co-crystallization with free ubiquitin revealed two additional binding sites—S2 and the UBD—that allow additional insights into DUB specificity and ubiquitin binding. The SARS virus DUB, PLpro, cleaves K48-poly-ubiquitin down to di-ubiquitin and also binds the I44 patch of Ub$_{S2}$ by utilizing a central Phe residue among other hydrophobic residues[42]. The biological relevance of stalling long-chain cleavage at di-ubiquitin—without further cleavage to mono-ubiquitin—warrants further investigation and may reflect a chain editing function of these DUBs.

The unique VR-1-equivalent segment of OtDUB includes a high-affinity UBD. Other bacterial DUBs, such as XopD and SseL, also encode UBDs, but in the form of a low-complexity region and a VHS domain, respectively[11]. Unlike these CE-clan proteases, the OtDUB VR harbors two separate ubiquitin-binding interfaces, both of which have been captured structurally. Interestingly, OtDUB$_{UBD}$:ubiquitin interaction strongly promotes DUB activity at high OtDUB concentrations. Saturation of the UBD with free ubiquitin both reduces competition for ubiquitylated substrates at the DUB active site and enhances S1 site formation by promoting folding of VR-1/UBD, which increases K48-chain recognition. Local subcellular concentrations of OtDUB during *Orientia* infection are not yet known, so it is

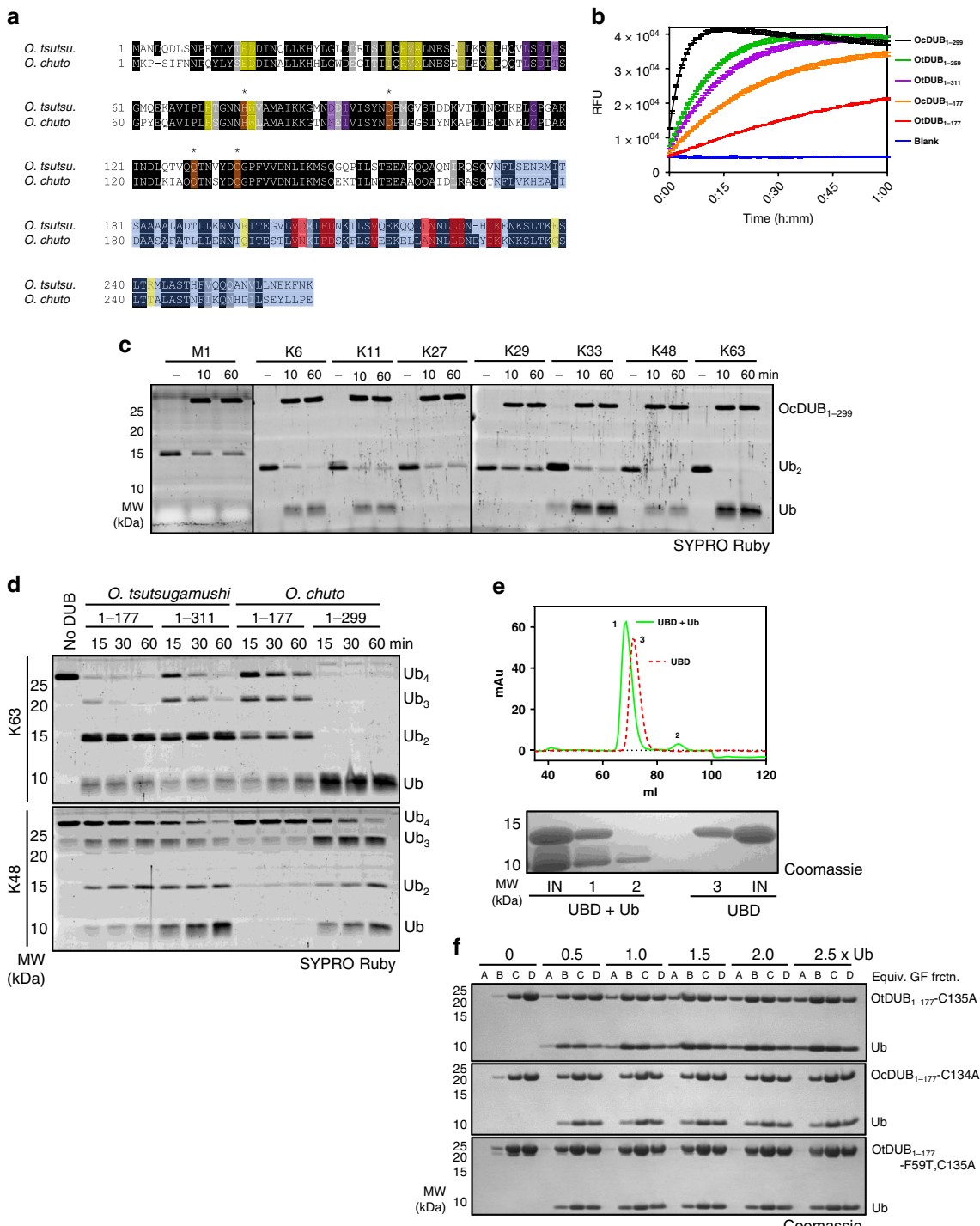

**Fig. 7 OtDUB activities are conserved in *Orientia chuto*. a** Amino acid alignment of the DUB/VR-1 region of OtDUB and its homolog in *Orientia chuto*. Identical and similar amino acids are boxed in black and gray, respectively, and the conserved catalytic triad and oxyanion hole residues are boxed in orange and starred. UBD boundaries are boxed in light blue. The ubiquitin-interacting residues are boxed for the S1 site (yellow), S2 site (purple) and the UBD (red). **b** Ub-AMC-cleavage assays comparing OtDUB and OcDUB activities. Ub-AMC (400 nM) was incubated alone (Blank) or with $OtDUB_{1-177}$, $OtDUB_{1-259}$, $OtDUB_{1-311}$, $OcDUB_{1-177}$, or $OcDUB_{1-299}$ (350 pM). Lines represent the means of technical triplicates with SD bars at each 40-s interval. **c** Representative results for an in vitro cleavage assay using di-ubiquitin substrates (1 μM each) that were incubated in the presence or absence of $OcDUB_{1-299}$ (0.5 μM) for the indicated times and resolved by SDS-PAGE (*n* = 2). Black lines denote separate gels. Source data are provided as a Source Data file. **d** In vitro cleavage of tetra-ubiquitin chains (2 μM) by $OtDUB_{1-177}$, $OtDUB_{1-311}$, $OcDUB_{1-177}$, or $OcDUB_{1-299}$ (50 nM) (*n* = 2). Source data are provided as a Source Data file. **e** SEC peak shift assay for *O. chuto* $UBD_{170-299}$-ubiquitin association (top). *Orientia chuto* $UBD_{170-299}$ of 2.5 mg were run on a Superdex 75 column alone (red), or mixed with equimolar amounts of ubiquitin (green), *n* = 2. Inputs and peak fractions were resolved by SDS-PAGE (bottom). Source data are provided as a Source Data file. **f** Ubiquitin-binding titration experiment for ubiquitin binding at the putative OcDUB S2 site. $OtDUB_{1-177}$ -C135A, $OcDUB_{1-177}$ -C134A, and $OtDUB_{1-177}$-F59T,C135A (1 mg, 250 μM) preincubated with the indicated molar ratio of ubiquitin were run on a Superdex 75 FPLC column. Equivalent fractions of each run (A, B, C, D) were resolved by SDS-PAGE. Source data are provided as a Source Data file.

unclear how relevant this is during infection. The high cytoplasmic levels of free ubiquitin may guarantee a fully activated DUB.

The OtDUB$_{UBD}$–ubiquitin interaction has unusual biophysical properties as well. Although the ubiquitin surface area buried by the bound OtDUB$_{UBD}$ (~17% of the ubiquitin surface) is virtually identical to that of one of the tightest UBDs with an available crystal structure, the CUE domain of Vps9, the binding affinity of the OtDUB$_{UBD}$ for ubiquitin is more than three orders of magnitude tighter[49]. Other properties of OtDUB$_{UBD}$ must therefore account for its tight association with ubiquitin. The OtDUB$_{UBD}$ utilizes a central hydrophobic interaction surface that contacts all three hydrophobic residues in the canonical Ile44 patch of ubiquitin[44] and is further stabilized by complementary electrostatic interactions surrounding the hydrophobic patch[38]. Shape complementarity (Sc)[50] may also help explain the tight affinity for ubiquitin. The calculated Sc value (obtained in CCP4) of the OtDUB$_{UBD}$:ubiquitin interface shows it to be one of the most complementary (0.65) interfaces among known UBDs; it is surpassed only by two other UBDs crystallized with ubiquitin, Rabex5 (0.69) and the Cbl-b UBA domain (0.72)[51,52]. While no single one of the aforementioned factors alone is sufficient, the combination of these and possibly other determinants may account for the exceptional affinity of the OtDUB$_{UBD}$ for ubiquitin, despite the entropic cost of UBD folding upon ubiquitin binding.

The OtDUB$_{UBD}$ will be useful as a tool for the enrichment and detection of ubiquitin and ubiquitylated proteins. Related to this, avidity-based, high-affinity protein binders specific for free ubiquitin have been engineered that are useful as sensors for free mono-ubiquitin levels[53]. Recent work has also shown that Cys-to-Ala mutation within some DUB catalytic triads results in high-affinity ubiquitin binding that could be used similarly[54]. To analyze OtDUB for this potential, we tested mono-ubiquitin binding by ITC to OtDUB$_{1-177}$ bearing a C135A mutation, but saw only weak binding ($K_d = 6.3 \pm 0.2\ \mu M$, $n = 1.8 \pm 0.04$) (Supplementary Fig. 7b). The stoichiometry of ~2 ubiquitins to one DUB is likely due to occupancy of the S1 and S2 sites.

In summary, we have uncovered a unique DUB from the obligate intracellular pathogen *O. tsutsugamushi* that is capable of efficiently cleaving long ubiquitin chains of different linkages and binding ubiquitin with extraordinary affinity. This UBD is structurally dynamic on its own, but folds into a unique structure in the presence of ubiquitin. The co-crystal structure of the DUB-UBD segment with ubiquitin revealed multiple ubiquitin-binding sites that can rationalize the specificity of the enzyme for longer ubiquitin chains and the high affinity of the UBD for ubiquitin. These characteristics are conserved in the *O. chuto* homolog, suggesting that this protein is critical for pathogenicity. These results both advance the range of effector proteins in these understudied pathogens that can be used for drug targeting and expand our understanding of the repertoire of ubiquitin binding and DUB mechanisms utilized in host–pathogen interactions.

## Methods

**Plasmids and cloning**. DNA plasmids used in this study can be found in Supplementary Data 1 along with primer sequences and molecular cloning strategies.

**Protein expression and purification**. For pGEX6P1-based plasmids, Rosetta DE3-transformed *Escherichia coli* were back diluted in Luria-Bertani (LB) or Terrific Broth with 100 μg/ml ampicillin, grown to an OD$_{600}$ 0.5–0.7, induced with 300 μM isopropyl β-D-1-thiogalactopyranoside (IPTG), and grown for 16 h at 18 °C. Bacteria was pelleted, resuspended in lysis buffer (phosphate-buffered saline (PBS) + 400 mM KCl, 1 mM dithiothreitol (DTT), 2 mM phenylmethylsulfonyl fluoride (PMSF), lysozyme, and DNase) and incubated for 1 h on ice prior to mechanical disruption by French pressing. Clarified lysates were incubated with glutathione resin (small scale; 1 ml resin/L culture rotated for 1 h at 4 °C, large scale; lysate

pumped at 1 ml/min over 20 ml resin bed). Resin was washed extensively with PBS + 400 mM KCl prior to elution with 3–5 CV (column bed volumes) of elution buffer (250 mM Tris-HCl pH 8, 0.5 M KCl, 10 mM reduced glutathione). When applicable, the N-terminal GST tag was cleaved with GST-HRV 3C during overnight dialysis in 2× 4 L of dialysis buffer (PBS + 400 mM KCl, 1 mM DTT). The cleaved GST and GST-HRV 3C were captured on glutathione resin and the flow through concentrated (Amicon Ultra) prior to fast protein liquid chromatography (FPLC) (Akta)-based SEC in 50 mM Tris-HCl, 150 mM NaCl, 1 mM DTT, or 50 mM HEPES pH 7.5, 100 mM NaCl (ITC buffer). Peak fractions were analyzed by Coomassie-stained SDS-PAGE, pooled, and concentrated. Protein concentrations were determined by extinction coefficient-adjusted $A_{260/280}$ values (Nanodrop) or by bicinchoninic acid assay (BCA). Proteins were flash frozen in liquid N$_2$ and stored at −80 °C.

For untagged ubiquitin, *E. coli* was grown and induced as above. The pelleted bacteria were resuspended in 50 mM NaOAc pH 4.5, 1 mM DTT, 2 mM PMSF, and lysozyme. French press lysates were then heated to 85 °C for 20 min, cooled to room temperature, and centrifuged. Clarified lysates were pumped over a 20 ml SP-Sepharose resin bed at 1 ml/min. The resin was washed with 5 CV of 50 mM NaOAc pH 4.5, 1 mM DTT, followed by 5 CV of 50 mM NaOAc pH 4.5, 50 mM NaCl, and 1 mM DTT. Ubiquitin was eluted with a NaCl step gradient of 100 mM increments (50 ml each) from 100 to 500 mM in 5 ml fractions. Ubiquitin-containing fractions were pooled, concentrated, and further purified by SEC FPLC on a Superdex 75 Hiload 16/600 pre-equilibrated with 50 mM Tris-HCl pH 7.5, 150 mM NaCl, and 1 mM DTT. Ubiquitin pure fractions were concentrated, BCA quantified, flash frozen in liquid N$_2$, and stored at −80 °C.

Bacteria transformed with pRT497-6His-MBP-hNEDD8 were out grown in LB + 50 μg/ml kanamycin and processed like pGEX6P1-based plasmids with the following exceptions: the lysis buffer was 50 mM Tris-HCl pH 8.0, 300 mM NaCl, 10 mM imidazole, 2 mM PMSF, lysozyme, and DNase. The clarified lysates were pumped over a 20 ml Ni-NTA resin bed (Qiagen), washed with 5 CV of 50 mM Tris-HCl pH 8.0, 300 mM NaCl, 10 mM imidazole, and 5 CV 50 mM Tris-HCl pH 8.0, 300 mM NaCl, 20 mM imidazole. Protein was eluted with 4 CV of 50 mM Tris-HCl pH 8.0 and 300 mM NaCl with 250 mM imidazole. The N-terminal 6His-MBP tag was cleaved with 6His-HRV 3C during overnight dialysis in 2× 4 L 50 mM Tris-HCl pH 8.0, 300 mM NaCl, and 1 mM DTT. The 6His-MBP and 6His-HRV 3C were captured on Ni-NTA resin and the flow through was concentrated prior to Superdex 75 SEC in 50 mM Tris-HCl pH 8.0 and 150 mM NaCl. For ITC experiments, the buffer was exchanged by overnight dialysis in 1 L of 50 mM HEPES pH 7.5 and 100 mM NaCl prior to the extensive dialysis done for ITC (see below).

For pET16b Cdc34, the bacteria were lysed in 20 mM NaPO$_4$ pH 7, 300 mM NaCl, lysozyme, and 1 mM PMSF. Clarified lysates were loaded onto an 20 ml Cobalt resin bed (HisPur/Thermo) and washed with 100 ml of 20 mM NaPO$_4$ pH 7, 300 mM NaCl, 5 mM imidazole and 100 ml of 20 mM NaPO$_4$ pH 7, 300 mM NaCl, 10 mM imidazole. Protein was eluted with 20 mM NaPO$_4$ pH 7, 300 mM NaCl, 200 mM imidazole, and further purified on a Superdex 75 HiLoad 16/600 pre-equilibrated with 50 mM Tris-HCl pH 7.5, 150 mM NaCl, and 1 mM DTT. Peak fractions were pooled, concentrated, flash frozen in liquid N$_2$, and stored at −80 °C.

For GST-pulldown proteins, DNA encoding a fusion protein of recombinant GST fused in frame to residues 170–259 of the *OTT_1962* ORF (open reading frame) were cloned into pET-28a with a tobacco etch virus (TEV) cleavage site as a linker. BL21(DE3) *E. coli* were transformed, grown, and induced as above. Clarified lysates were applied to Ni-NTA agarose, washed extensively, and eluted with buffer containing 250 mM imidazole. Elution fractions from Ni-NTA were directly applied to a 5-ml GSTrap HP (GE Healthcare) column at 1 ml/min using an ÄKTA FPLC (GE Healthcare), washed extensively, and eluted with 10 mM reduced glutathione. GST elution fractions were pooled, concentrated, and dialyzed into 25 mM Tris-HCl pH 8.0, 100 mM NaCl, and 0.1 mM TCEP to remove bound glutathione.

DNA sequences for OtDUB$_{1-259}$ and His-tagged human ubiquitin were cloned into a modified pET-28a vector encoding a TEV-cleavable hexahistidine tag at the N terminus. Following protein purification and TEV cleavage, a single nonnative glycine remained at the N terminus of OtDUB$_{1-259}$ and a tri-glycine upstream of ubiquitin. BL21(DE3) *E. coli* were grown in LB overnight, back diluted in Terrific Broth supplemented with 40 μg/ml kanamycin to OD$_{600}$ 0.6–0.8 at 37 °C, induced with 500 μM IPTG, and grown for 16 h at 18 °C. Pelleted bacteria were resuspended in 50 mM Tris-HCl pH 8.0, 500 mM NaCl, 0.1 mM tris(2-carboxyethyl)phosphine (TCEP) supplemented with protease inhibitors (Roche). Clarified lysates were incubated with 10 ml Ni-NTA agarose resin (Qiagen) for 30 min, washed with 15 CV of lysis buffer, and eluted with lysis buffer supplemented with 250 mM imidazole pH 8.0. Pooled elution fractions were concentrated, mixed with His-tagged TEV protease (1:100 mass ratio), and simultaneously dialyzed into their respective ion exchange buffers: 50 mM Tris-HCl pH 8.0, 0.1 mM TCEP for OtDUB; 50 mM sodium acetate pH 4.0, 0.1 mM TCEP for ubiquitin. After 72 h, proteins were subjected to ion exchange chromatography, both from a linear gradient of 0–1 M NaCl:HiTrap Q for OtDUB and HiTrap S for ubiquitin. Peak fractions were subjected to SDS-PAGE analysis, pooled, concentrated, and separately injected onto a HiLoad Superdex 75 pg column (GE Healthcare) equilibrated with 25 mM Tris-HCl pH 7.5, 100 mM NaCl, 0.1 mM TCEP. For complex formation, the two proteins were mixed in a 1:5::OtDUB:Ub ratio, and re-

injected onto the gel filtration column; complex fractions were identified by SDS-PAGE, pooled, concentrated to ~50 mg/ml, flash frozen in liquid $N_2$, and stored at −80 °C.

**Selenomethionine labeling**. pGEX6P1-OtDUB$_{1-311}$-transformed *E. coli* were back diluted and grown in M9 minimal media + 100 μg/ml ampicillin to an OD$_{600}$ 0.5–0.6. Cultures were spiked with a mixture of powdered amino acids (100 mg lysine, 100 mg phenylalanine, 100 mg threonine, 50 mg isoleucine, 50 mg leucine, 60 mg selenomethionine (SeMet); Cayman Chemical Company), shifted to 18 °C for 15 min and then induced with 300 μM IPTG for 16 h. Purification was the same as pGEX6P1-based plasmids, except 4 mM DTT was used in all buffers.

**Ubiquitin and UBD labeling for NMR**. Transformed *E. coli* were grown in M9 minimal medium + 100 μg/ml ampicillin supplemented with MEM vitamins (Thermo). For ubiquitin, the medium contained $^{15}$NH$_4$Cl, and for the UBD, the medium had $^{15}$NH$_4$Cl and $^{13}$C glucose (Cambridge Isotope Laboratories). Bacteria were grown to an OD$_{600}$ 0.6–0.7, induced with 300 μM IPTG, and grown for 16 h at 18 °C. Both were purified as above using pGEX6P1-based vectors.

**OtDUB$_{1-311}$ apo protein crystallization and data collection**. Crystallization screening of OtDUB$_{1-311}$ was performed by sitting drop vapor diffusion at room temperature. Diffraction data were obtained at the Advanced Photon Source (APS), beamline 23-ID-D, from a crystal that grew in a solution of 15 mg/ml OtDUB$_{1-311}$ and mother liquor containing 0.2 M ammonium iodide and 20% polyethylene glycol (PEG) 3350. The crystal was flash frozen in liquid nitrogen after cryo-protection with 25% ethylene glycol and diffracted to 1.74 Å. Molecular replacement was unsuccessful; therefore, we purified OtDUB$_{1-311}$ with SeMet labeling (see below). Crystals were grown overnight by sitting drop vapor diffusion from 15 mg/ml protein in the same mother liquor as before, cryo-protected in 25% ethylene glycol, and flash frozen in liquid nitrogen. Diffraction data were collected at the APS on beamline 24-ID-E at 100 K temperature using wavelength 0.98 Å, and crystals were diffracted to 2.0 Å. Data statistics are summarized in Table 2.

Diffraction data were processed in HKL2000[55] in space group $P2_12_12$. Initial phases for the SeMet derivative data were obtained using the single anomalous dispersion (SAD) method. The selenium substructure was solved experimentally in Shelx[56] and subsequent SAD phasing was performed in SOLVE[57]. A single copy of the OtDUB fragment was identified in the electron density map. Iterative rounds of model building in Coot[58] and refinement with Refmac5[59] were performed. Evaluation of the Ramachandran plot gave 95.9% in favored, 3.7% in allowed, and 0.4% in outlier regions. Refinement statistics are summarized in Table 2.

**OtDUB$_{1-259}$/ubiquitin complex crystallization and data collection**. For the crystallization of OtDUB$_{1-259}$/ubiquitin, the protein complex was diluted to ~30 mg/ml and mixed with commercial crystallization formulations using the micro-batch under oil method[60] at room temperature. Several dozen crystallization hits were identified after <24 h; large, rod-shaped crystals formed in conditions containing potassium thiocyanate (KSCN) and moderate concentrations (20–30%) of low molecular-weight PEGs. The final crystal was formed by streak seeding a drop composed of 2 μl of protein complex mixed with 2 μl of 150 mM KSCN and 32.5% (w/v) PEG 3350. Crystals were partially cryo-protected in situ by the addition of crystallization buffer supplemented with 25% (v/v) glycerol and flash frozen in liquid nitrogen. Diffraction data were collected at the APS on beamline 24-ID-E at 100 K temperature and wavelength 0.98 Å. The crystal diffracted to 2.2 Å resolution. Data collection statistics are summarized in Table 2.

Diffraction data were processed with HKL2000[55] in space group $P2_12_12_1$ and molecular replacement was performed using PHASER[61] with the previously solved OtDUB$_{1-311}$ apo structure. Initially, no solution was found by PHASER; therefore, the apo model was split into two domains: the DUB (1–169) and VR-1/UBD (171–259), and an initial solution was found with two copies of each fragment per asymmetric unit (asu). Using the OtDUB$_{1-259}$ model with domain conformations from the initial search, two additional copies of OtDUB$_{1-259}$ were found within the asu. The initial model was first improved with several rounds of restrained refinement in Refmac5[59], and additional rounds of molecular replacement were attempted with human ubiquitin (PDB ID: 1UBQ) as the search model. Unexpectedly, PHASER was unable to find any copies of ubiquitin within the unit cell despite extensive difference density and readily identifiable secondary structure. Therefore, individual molecules of ubiquitin were manually docked into the electron density in Coot, and iterative rounds of restrained refinement improved phases such that sidechain density for all three unique molecules of ubiquitin emerged. In total, four molecules of the OtDUB fragment and 12 molecules of ubiquitin were placed within the asu. The model was rebuilt in Coot according to the $2F_o–F_c$ map, followed by iterative rounds of restrained, translation/libration/screw and twin refinement, with twin operator (−h, −k, −l) and twin fraction $\alpha = 0.26$. Evaluation of the Ramachandran plot gave 97.9% in favored, 2.0% in allowed, and 0.1% in outlier regions. Refinement statistics are summarized in Table 2.

**NMR data collection**. The $^1$H-$^{15}$N HSQC (heteronuclear single quantum coherence) spectrum for apo UBD was acquired on a 600 MHz Varian Inova

### Table 2 Data collection and refinement statistics.

| | OtDUB$_{1-311}$ SeMet | OtDUB$_{1-259}$:Ub |
|---|---|---|
| **Data collection** | | |
| Space group | $P2_12_12$ | $P2_12_12_1$ |
| Cell dimensions | | |
| $a$, $b$, $c$ (Å) | 114.79, 43.09, 58.24 | 119.75, 144.07, 143.47 |
| $\alpha$, $\beta$, $\gamma$ (°) | 90, 90, 90 | 90, 90, 90 |
| Resolution (Å) | 2.0 (2.07–2.00)ᵃ | 2.2 (2.28-2.20) |
| $R_{sym}$ or $R_{merge}$ | 0.144 (0.825) | 0.081 (0.909) |
| $I$ /σ$I$ | 10.6 (1.6) | 10.6 (1.1) |
| Completeness (%) | 91.0 (93.7) | 98.8 (94.2) |
| $CC_{1/2}$ | 0.990 (0.702) | 0.998 (0.318) |
| Redundancy | 4.3 (4.0) | 4.6 (4.3) |
| **Refinement** | | |
| Resolution (Å) | 50.0–2.00 | 56.6-2.20 |
| No. of reflections | 18,794 (1895) | 124,050 (11,681) |
| $R_{work}$/$R_{free}$ | 0.20/0.24 (0.34/0.36) | 0.17/0.21 (0.18/0.21) |
| No. of atoms | | |
| Protein | 1926 | 15,355 |
| Ligand/ion | – | – |
| Water | 80 | 477 |
| *B*-factors (Å²) | | |
| Protein | 27.8 | 26.4 |
| Ligand/ion | – | — |
| Water | 42.9 | 39.9 |
| R.m.s. deviations | | |
| Bond lengths (Å) | 0.015 | 0.018 |
| Bond angles (°) | 1.64 | 1.97 |

One crystal for each dataset was used for data collection and structure determination.
ᵃStatistics for the highest-resolution shell are shown within parentheses.

spectrometer equipped with a triple resonance probe and pulsed field gradients. The spectra were acquired with 16 scans, 64 t1 increments in the $^{15}$N dimension, and 12k Hz and 2500 Hz spectral widths in the $^1$H and $^{15}$N dimensions, respectively. The spectra for the 1:0.5 and the 1:1 complex of UBD with ubiquitin were acquired with identical parameters, except the nitrogen dimension was acquired using 128 t1 increments. The acquired spectra were processed identically in NMRPipe[62] and analyzed with Sparky[63].

**UBL-AMC and ubiquitin cleavage assays**. AMC-cleavage experiments were carried out as previously described[64]. Ub-AMC, NEDD8-AMC, ISG15-AMC, and SUMO1-AMC (Boston Biochem) were diluted in AMC-cleavage buffer (50 mM Tris-HCl, pH 7.5, 500 μM EDTA, 5 mM DTT, 0.1 % (w/v) bovine serum albumin), added (60 μl at 666 nM) to a 96-well black polystyrene plate (Costar) and equilibrated by shaking at 30 °C for 5 min in a fluorescent plate reader (Synergy MX, BioTek). OtDUB$_{1-259}$ diluted to the appropriate concentration was quickly added (40 μl of 8.75 nM or 875 pM) to each well, mixed for 15 s by shaking, and then datapoints were collected at 30 °C, every 40 s for 60 min by 345/445 nm excitation/emission.

DUB activity was analyzed using a published ubiquitin cleavage assay[65] where the described concentrations of ubiquitin chains and DUB protein fragments were incubated at room temperature in 50 mM Tris-HCl pH 7.5, 20 mM KCl, 5 mM MgCl$_2$, and 1 mM DTT for the indicated times before being quenched with Laemmli sample buffer and placed on ice. Reactions were then resolved on 15% SDS-PAGE gels and detected by SYPRO Ruby staining or ubiquitin immunoblotting. SYPRO Ruby-stained gels were imaged on a Bio-Rad ChemiDoc and quantified using ImageJ[66]. Images were processed in Adobe Photoshop by inverting, uniformly adjusting levels, and cropping.

Supplementary Fig. 7c is a Coomassie-stained SDS-PAGE gel highlighting the purity of the enzyme variants used in activity assays.

**K48-linked ubiquitin chains**. Extended K48 ubiquitin chains were generated in an overnight room temperature reaction containing: human UBE1, Cdc34, an excess of WT ubiquitin in an ATP regenerating system (50 mM Tris-HCl pH 8.0, 5 mM MgCl$_2$, 10 mM creatine phosphate, 0.6 U/ml inorganic pyrophosphatase, 0.6 U/ml creatine phosphokinase, 2.5 mM ATP, and 0.5 mM DTT)[67]. Due to the poor activity of Cdc34 in vitro, the majority of assembled chains were between di- and tetra-ubiquitin, with small amounts of penta-ubiquitin and larger chains. The reaction was quenched by diluting with a 10-fold excess of 50 mM NaOAc pH 4.5, filtered, and loaded onto a cation exchange chromatography column (Mono S).

Di-, tri-, and tetra-ubiquitin eluted as individual peaks across a 0–250 mM NaCl gradient. Pooled peak fractions were buffer exchanged (50 mM Tris-HCl pH 7.5, 150 mM NaCl) and concentrated in centrifuge filters (Amicon Ultra, 3 K cut off). Concentrated proteins were quantified by extinction coefficient-adjusted $A_{260/280}$ values (Nanodrop), aliquoted, flash frozen in liquid $N_2$, and stored at −80 °C.

**K63 di-ubiquitin**. To synthesize K63 di-ubiquitin excess ubiquitin K63R and ubiquitin D77 were incubated at room temperature overnight with human UBE1, Uev1a, and Ubc13 in an ATP regenerating system. The di-ubiquitin was isolated from residual mono-ubiquitin and reaction enzymes by cation exchange chromatography (see above). Concentrated di-ubiquitin fractions were further purified on a Superdex 75 Hiload 16/600 equilibrated with 50 mM Tris-HCl pH 7.5, 150 mM NaCl. Peak fractions were concentrated, and then quantified by extinction coefficient-adjusted $A_{260/280}$ values (Nanodrop), aliquoted, flash frozen in liquid $N_2$, and stored at −80 °C.

**K63 tri- and tetra-ubiquitin**. K63 di-ubiquitin was de-blocked with UCLH3 for 1 h at 37 °C in 50 mM Tris-HCl, 1 mM EDTA, and 1 mM DTT. The reaction was quenched with excess NaOAc pH 4.5 and isolated by cation exchange chromatography[66]. De-blocked K63 di-ubiquitin was then used as substrate to generate extended K63 chains in an overnight room temperature reaction with human UBE1, Uev1a, and Ubc13 in an ATP regenerating system. Reactions were then processed as the K48 extended chains above.

**GST-pulldown assays**. Purified proteins were assayed for binding by mixing GST-tagged UBD (0.2 mg; 25 μM) with His-tagged ubiquitin (0.2 mg; 75 μM) in binding buffer (25 mM Tris-HCl pH 8.0, 100 mM NaCl, 0.1 mM TCEP) to a final volume of 200 μl. Protein mixtures were applied to pre-equilibrated glutathione sepharose 4B resin (GE Healthcare), mixed thoroughly, and incubated for 1 h at 2 °C. The resin was washed with 15 CV of binding buffer and proteins eluted with 4 CV of binding buffer supplemented with 10 mM reduced glutathione. Eluates were analyzed by SDS-PAGE and stained with Coomassie blue. FOTO/Analyst PC Image was used for collecting Coomassie-stained gel images.

**Isothermal titration calorimetry**. Purified proteins were pre-equilibrated for ITC in 50 mM HEPES pH 7.5, and 100 mM NaCl by extensive dialysis (4 × 500 ml) over 2 days at 4 °C using Slide-A-Lyzer MINI Dialysis Units 3500 MWCO (Thermo). After dialysis, protein concentrations were determined by BCA and the last volume of dialysis buffer was retained for diluting the proteins and equilibrating the ITC chamber/syringe. ITC experiments were carried out in a Nano ITC apparatus (TA Instruments) using the following parameters: 50 μl of protein in the syringe (rotating at 225 RPM) was injected in 31 total injections (1 = 0.75 μl, 2–31 = 1.6 μl) with 180-s intervals into 310 μl of protein in the chamber (190 μl working volume). Experiments were analyzed using NanoAnalyze (TA Instruments).

**Peak shift assays**. The described concentration of each protein was mixed or diluted alone in a 500 or 600 μl reaction and incubated for 30 min at 37 °C. Potential aggregates were pelleted at 21,000 × g for 5 min prior to loading on the Superdex 75 HiLoad 16/600 SEC. All peak shift assays were carried out in 50 mM HEPES pH 7.5, 100 mM NaCl, except for the NEDD8 experiments, which were done in 50 mM Tris-HCl pH 8, 150 mM NaCl. Input and peak fractions were resolved by SDS-PAGE and Coomassie stained (Gel Code Blue) and captured using a G:Box imaging system with GeneSnap software (Syngene).

For titration experiments, 1 mg (250 μM) of OtDUB or OcDUB was mixed with the described molar equivalents of ubiquitin (125–750 μM) in a final volume of 200 μl, incubated for 30 min at 25 °C, and aggregates pelleted at 21,000 × g for 5 min prior to loading on the Superdex 75 10/300 GL SEC. Samples were eluted at a constant rate of 0.5 ml/min in 50 mM HEPES pH 7.5, 100 mM NaCl. Peak fractions were resolved by SDS-PAGE and Coomassie stained and captured using FOTO/Analyst PC Image.

**Immunoblotting**. SDS-PAGE gels were transferred to Immobilon-P PVDF (Millipore), blocked 5% milk (w/v) in Tris-buffered saline with 0.1% Tween-20 (TBST) incubated with a primary antibody in 5% milk in TBST (polyclonal rabbit anti-ubiquitin antibody [Z0458, Dako] at 1:2000), followed by incubation with a peroxidase-coupled anti-rabbit-IgG antibody (NA934V, GE Healthcare), 1:5000 or 1:10,000 (v/v) in 5% milk in TBST. Blots were visualized by enhanced chemiluminescence[68] on film (Denville) or a G:Box imaging system with the GeneSnap software (Syngene). Images were processed in Adobe Photoshop by uniformly adjusting levels and cropping.

**Statistics**. Unpaired, two-tailed $t$ tests were performed for all statistical analyses. Reported $p$ value significance in each figure are as follows: *<0.05, **<0.005, ***<0.0005, and n.s., not significant. For exact $p$ values and additional statistical parameters refer to the source data file. Measurements were taken from distinct samples of independent cleavage assays.

**Accession numbers**. Coordinates and structure factors have been deposited in the Protein Data Bank with PDB ID: 6UPS (SeMet apo) and 6UPU (native complex).

**Reporting summary**. Further information on research design is available in the Nature Research Reporting Summary linked to this article.

## Data availability
Source data for all relevant figures are supplied in the Source Data File along with statistical analysis parameters. This includes: Figs. 1e, 3c–d, f–g, 4b–c, e–f, 5b, d, 6a–e, g–h, 7c–f, and Supplementary S1a–c, S1a, S2c, S5b, S6a–f, S7c. PDB ID: 6UPS (SeMet apo) and 6UPU (native complex).

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

## Acknowledgements

We thank Cynthia Wolberger (Johns Hopkins University School of Medicine) for the pET SUMO2 UBCH5A plasmid[69], Chittaranjan Das (Purdue University) for plasmids required for in vitro ubiquitin-chain synthesis[65], Tetsuya Hayashi (U. of Miyazaki, Japan) for *O. tsutsugamushi* Ikeda DNA and Ulrike Munderloh (U. of Minnesota) for *R. bellii* DNA. This work was supported by NIH Grants R01 GM046904 and R01 GM053756 (to M.H.), R01 AI116313 (to Y.X.), and R01 GM106121 (to J.P.L.). Additional training support was provided by a NIH Fellowship F32 GM113456 (to J.M.B.) and a National Science Foundation Fellowship DGE1122492 (to C.L.). We are grateful to our host Craig Ogata at the GM/CA CAT beamline 23-ID-D as well as Raj Rajashankar and Sukumar Narayanasami at the NE-CAT 24-ID-E beamline at the Advanced Photon Source of Argonne National Laboratory. This work was supported by the following: NE-CAT beamlines (GM103403) and a Pilatus detector (RR029205) at the APS (DE-AC02-06CH11357).

## Author contributions

M.H., J.F.B., and J.A.R. conceived and initiated the project. J.M.B., C.L., and J.A.R. designed, performed, and analyzed the experiments. A.C. carried out NMR experiments and analyzed them with J.P.L. H.C. performed experiments. Y.X. assisted C.L. and J.A.R. in solving and refining the crystal structures. J.M.B., C.L., and M.H. wrote the manuscript. All authors provided critical feedback on the manuscript.

## Competing interests

M.H. and J.M.B. are inventors on a US Provisional Patent Application No. 62/909,545 filed on 2 October 2019 that covers methods of ubiquitin detection and enrichment using the OtDUBUBD. The other authors declare no competing interests.
