## [Peer Review File · Nature Communications]

Reviewers' comments:

Reviewer #1 (Remarks to the Author):

NCOMMS-19-37565

Review comment:

Berk et al., "A Deubiquitylase with an Unusually High-affinity Ubiquitin-binding Domain from the Scrub Typhus Pathogen *Orientia tsutsugamushi*"

Effector proteins of pathogenic bacteria can disturb and/or hijack immune signaling pathways of host cells. Such signaling pathways are controlled by ubiquitylation, and therefore, some effector proteins possess E3 ubiquitin (Ub) ligase activity or deubiquitylating (DUB) activity. The CE clan/Ulp1-like proteases are a major DUB family of bacterial effectors. Functional and structural mechanisms underlying their DUB activities have been extensively studied. In this study, Berk et al. present the apo- and Ub-bound structures of the CE clan/Ulp1-like protease domain in a likely effector protein of the obligate intracellular bacterium *Orientia tsutsugamushi* (OtDUB). OtDUB binds to three Ub molecules, which occupy the S1 and S2 sites in the catalytic domain and the high-affinity binding site in the additional Ub-binding domain (UBD). The S2 site or the additional UBD is not included in the CE clan/Ulp1-like protease domains of known structures. However, unfortunately, physiological significance of these two structural properties is unclear. The structure suggests the preference to K63 linkage, but OtDUB seems to cleave K48-linked chains as well as K63-linked chains. The functional role of the additional UBD is also unclear. I would like to suggest the following points to be improved if this study would be published.

Major points:

1. Orders of figures are inconsistent (e.g., Fig. 1c is cited before Fig. 1b).
2. In Fig. 1a, VR-2, CR, and VR-3 should be indicated. Also, the color of VR-3 is indistinguishable.
3. Fig. 1c is difficult to interpret. Please align the standard cartoon models with the conserved CE clan fold in similar orientations. Do not superpose the structures.
4. It would be better if tetra M1- and K11-linked Ub chains could be tested as substrates for OtDUB.
5. The difference between the apo- and Ub-bound OtDUB structures should be mentioned.
6. In Fig. 2b, please compare the OtDUB-Ub(S1) and other bacterial DUB-Ub/Ubl(S1) structures with Ub(S1) in similar orientations.
7. In Fig. 3b, Ile44 of Ub(S2) should be indicated.

8. In Fig. 3c, K48-linked diubiquitin seems to be cleaved. Please explain this. Other data show that OtDUB cleaves triubiquitin or longer chains.
9. In Fig. 6a, please indicate the positions of the S1 and S2 sites and the UBD.
10. In Fig. 6b and Supplementary Fig. 1c, the cleavage of K48-linked Ub chains by *O. chuto* DUB (1-299) and *O. tsutsu* DUB (1-177) should be tested.
11. Phe59 of OtDUB is replaced by Thr in *O. chuto* DUB (Fig. 6a). In Fig. 3c, the F59T mutation of OtDUB decreases the cleavage activity for K63-linked chains. However, in Fig. 6b, *O. chuto* DUB shows substantial activity for K63-linked chains. Does *O. chuto* DUB really have the S2 site? Please explain this.
12. Is pre-incubation performed in Fig. S1b also effective for UBD mutants (V203D, F207D and L221D)?
13. In Fig. S2, please compare OtDUB-Ub with OTUB1-Ub, where two Ub molecules are aligned so as to form a K48 linkage.
14. In Methods, twin refinement is mentioned. Twinning parameters should be shown.

Minor points:

1. The terms "OTT_1962" and "OtDUB" are used in the figures. Please use either one.
2. In Fig. 2, the position of Lys48 should be indicated.
3. "Supplemental Figure" may be "Supplementary Figures" in Nature style.
4. In Methods, "DNAse" should be "DNase".

Reviewer #2 (Remarks to the Author):

In their submitted manuscript, Berk & Lim et al. present a thorough structural and biochemical analysis of a novel deubiquitinase domain from the scrub typhus pathogen *Orientia tsutsugamushi*. This work is a fascinating addition to our understanding of a growing family of bacterial deubiquitinases from the CE clan of proteases and describes several discoveries that are thus far unique to OtDUB and related *Orientia* homologs. In particular, the preference for longer polyubiquitin chains and surprising high-affinity interaction with monomeric ubiquitin are very striking and new to this family of deubiquitinases. The work is of high quality and will be of broad interest to researchers studying the ubiquitin system or host-pathogen interactions. Barring several

comments outlined below, I find this manuscript to be well-suited for publication in Nature Communications.

1) The authors seem to explain the lack of activity against di-ubiquitin chains as resulting from a requirement for longer (K63-linked) chains binding into the S2 site. If this is true, then why is OtDUB also more active against longer K48-linked chains, when based on the structure the S1-S2 connectivity appears to be K63-specific?

2) On a related note, why is it that K48 chains can be cleaved to produce monoUb, while K63 chains cannot (see Fig. 1e-f)? And why is the enzyme active against the simple Ub-AMC substrate?

3) Considering the above comments, I am curious if the active site (S1'-S1 connection) is incompatible with K63-linked chains, while other linkages (e.g. K48 or Ub-AMC) can be cleaved. Have the authors considered the possibility that OtDUB targets hybrid chains?

4) In Figure 4, the authors present mutations along one face of VR-1 that target the S1 site. These mutations appear to affect cleavage of K48 chains but not K63 chains, presumably because the effect on K63-linked chains is masked by the additional S2 interaction. Can the authors support this with a Ub-AMC assay? And possibly with S1-site mutations in the VR-2 region as well?

5) Given the disorder-order transition observed in VR-1 upon Ub binding (Fig. 5f), I am surprised that blocking this interaction with the V203D mutation does not maintain the disordered state enough to affect Ub binding to the S1 site and thus DUB activity (Fig. S4a). By eye this mutant might be slightly slower than the WT enzyme. Quantitation of this assay and/or kinetic analysis with the Ub-AMC substrate could be more telling and possibly indicate a role for the OtDUB VR-1 UBD in activating DUB function (akin to the interesting regulation of OTUB1 in humans).

Minor comments:

1) The OcDUB appears to be more active against di-ubiquitin substrates. Have the authors considered performing the full di-ubiquitin cleavage panel with this enzyme to profile specificity across the active site?

2) Without experimentation, claims that the UBD is specific for mono-Ub over chains are perhaps overstated.

3) In future assays, the "time zero" samples would be more representative if the individual reaction components were mixed directly into SDS sample buffer.

4) The authors might consider discussing the tUI ubiquitin sensor published by the Cohen lab as a higher affinity (though artificial) ubiquitin binder.

5) Please include CC1/2 values in Table 1, or otherwise justify why such an aggressive I/sigI cutoff was acceptable.

6) Why did the authors choose to not use the higher resolution native dataset for apo OtDUB?

Sincerely,

Jonathan Pruneda

Reviewer #3 (Remarks to the Author):

Berk et al. reported the structural characterization and ubiquitin chain linkage specificity of a deubiquitylase from *Orientia tsutsugamushi*. The crystal structures of the DUB domain of OtDUB alone and bound to three molecules of ubiquitin were presented. The structure of OtDUB reveals a similar core fold of CE clan proteases with distinct variable regions that are involved in ubiquitin binding. Surprisingly when co-crystallizing OtDUB with free ubiquitin, there were three ubiquitin bound in the structure. One bound at the S1 site, one at the S2 site and the other bound to a unique backside surface of the C-terminal region of OtDUB which the authors referred to as the ubiquitin-binding domain (UBD). Interestingly the C-terminus of ubiquitin at S2 is in proximity of K63 of ubiquitin at S1 leading to the hypothesis of K63-ubiquitin chain ($n>3$) recognition. The authors showed that OtDUB has preference for cleaving longer K63 ubiquitin chain ($n>2$) and does not cleave di-ubiquitin. Likewise it also cleaves K48-ubiquitin chain. The structure explains a model for how OtDUB could bind a tri-K63-ubiquitin. Mutation in the S2 binding site reduced cleavage of K63-tetra-ubiquitin but had little effect on K48-tetra-ubiquitin. While it is unclear how OtDUB recognizes K48-ubiquitin chain, the structure showed that K48 of S1 ubiquitin is accessible for conjugation with ubiquitin. Lastly the authors showed that the backside of UBD has a strong binding affinity for ubiquitin (K_d of 5 nM), which is two-orders of magnitude tighter than previously reported UBD-ubiquitin interaction.

This work reports a novel structure of OtDUB in complex with ubiquitin at S1 and S2 sites and explains how it recognizes K63-ubiquitin chain. In addition, it identifies a high-affinity UBD. These findings will be of interests to researchers in the ubiquitin field in understanding ubiquitin recognition and chain specificity. There are few concerns see below.

Comments

1. It is a bit confusing that in some assays 1-259 (Figure 3c, 4b, 6b, S4) was used and in others 1-311 was used (Figure 1d-f, S1B). Do both constructs have similar activity? It seems that 1-259 was used when comparing with mutants. Please explain.

2. This reviewer assumes that the same amount of OtDUB was used in the assay for WT and mutant. Please show a SDS-PAGE of OtDUB WT and mutants used in this study to illustrate their purity.

3. In the supplementary text and supplementary Figure 1B, the authors showed that in the absence of added free ubiquitin, OtDUB failed to cleave K63-tri-ubiquitin completely due to product inhibition. Is free ubiquitin added in all assays in other figures? It seems that K63-tetra-ubiquitin (Fig. 3b, 4b versus supplementary Figure 1B) is less susceptible to inhibition (by 10 min nearly all tetra-ubiquitin is cleaved). Is this due to tri vs tetra-ubiquitin? Is this specific to K63 or the same for K48-ubiquitin chain?

The authors stated in the supplementary text that 1-177 (Fig. S1B) is more efficient in cleaving K63-tetra-ubiquitin compared to 1-311 (Fig. 1E) suggesting that UBD could compete for ubiquitin binding thereby inhibiting the cleavage. First, these reactions were performed on separate western blots and it should be performed on the same gel to make such comparison. Second, Fig S4A showed that V203D abrogated UBD-ubiquitin interaction and exhibited similar activity as WT. If UBD has an inhibitory role one would expect V203D to be more active. Lastly, Fig.4A right panel showed that VR-1 makes extensive contact with S1 ubiquitin and 1-177 construct lacks this whole surface. One would predict to have a more severe defect than the point mutations in Fig. 4B and could impact on the cleavage of K63 chain. Please explain.

It would be useful to speculate the role of UBD-ubiquitin interaction in the discussion.

4. In Figure 3C, it seems that K63-tetra-ubiquitin is cleaved to di-ubiquitin whereas K48-tetra-ubiquitin is cleaved to mono-ubiquitin. Please discuss this difference. Also in Figure S1B, K48-di-ubiquitin was not cleaved at all, but when tetra-ubiquitin was used even di-ubiquitin was cleaved. Please explain.

5. Figure 1D legend missing. Please specify the concentration of Ubl-AMC or 1X

6. p.16 top "Instead the backside of the UBD domain (VR-1) is necessary for optimal K48-chain cleavage (Figure 4B and 4C)". Throughout the manuscript backside was referred to UBD-ubiquitin interaction. Please revise to S1 binding site.

We would like to thank all the reviewers for their constructive comments. The revisions that resulted have definitely improved the paper and strengthened its conclusions.

Reviewers' comments:

Reviewer #1 (Remarks to the Author):

NCOMMS-19-37565

Review comment:

Berk et al., "A Deubiquitylase with an Unusually High-affinity Ubiquitin-binding Domain from the Scrub Typhus Pathogen *Orientia tsutsugamushi*"

Effector proteins of pathogenic bacteria can disturb and/or hijack immune signaling pathways of host cells. Such signaling pathways are controlled by ubiquitylation, and therefore, some effector proteins possess E3 ubiquitin (Ub) ligase activity or deubiquitylating (DUB) activity. The CE clan/Ulp1-like proteases are a major DUB family of bacterial effectors. Functional and structural mechanisms underlying their DUB activities have been extensively studied. In this study, Berk et al. present the apo- and Ub-bound structures of the CE clan/Ulp1-like protease domain in a likely effector protein of the obligate intracellular bacterium *Orientia tsutsugamushi* (OtDUB). OtDUB binds to three Ub molecules, which occupy the S1 and S2 sites in the catalytic domain and the high-affinity binding site in the additional Ub-binding domain (UBD). The S2 site or the additional UBD is not included in the CE clan/Ulp1-like protease domains of known structures. However, unfortunately, physiological significance of these two structural properties is unclear. The structure suggests the preference to K63 linkage, but OtDUB seems to cleave K48-linked chains as well as K63-linked chains. The functional role of the additional UBD is also unclear. I would like to suggest the following points to be improved if this study would be published.

Major points:

1. Orders of figures are inconsistent (e.g., Fig. 1c is cited before Fig. 1b).

The order has been corrected.

2. In Fig. 1a, VR-2, CR, and VR-3 should be indicated. Also, the color of VR-3 is indistinguishable.

Variable regions have now been indicated.

3. Fig. 1c is difficult to interpret. Please align the standard cartoon models with the conserved CE clan fold in similar orientations. Do not superpose the structures.

Variable regions have been indicated. Thank you for the suggestion. Fig 1c (new Fig 1b) has been clarified according to the reviewer's comment.

4. It would be better if tetra M1- and K11-linked Ub chains could be tested as substrates for OtDUB.

In considering the comments from all reviewers about Ub-AMC being a suitable substrate but the OtDUB not detectably cleaving any diUb as assayed by anti-ubiquitin western blot, we decided to retest the diUb panel with SYPRO Ruby stain as the readout. In fact, we see cleavage of not only K48 and K63 chains, but also K6, K11 and K33 chains. M1 chains were not cleaved. See new Fig 1e.

5. The difference between the apo- and Ub-bound OtDUB structures should be mentioned.

RMSD difference between apo and Ub-bound structures has been added.

6. In Fig. 2b, please compare the OtDUB-Ub(S1) and other bacterial DUB-Ub/Ubl(S1) structures with Ub(S1) in similar orientations.

Fig 2c has been added to compare bacterial DUBs with S1 Ub/Ubl included.

7. In Fig. 3b, Ile44 of Ub(S2) should be indicated.

Ile44 has been indicated on S2 Ub

8. In Fig. 3c, K48-linked diubiquitin seems to be cleaved. Please explain this. Other data show that OtDUB cleaves triubiquitin or longer chains.

In light of all our new data examining diUb cleavage (Figures 1e, 3f, 4e, 6a,c,e). K48 diUb cleavage in 3c now makes more sense. Based on the enzymes concentrations and time points used, the cleavage of extended chains (trimers and dimers) is still significantly faster than diUb. Comparing 3c and 3f for example.

9. In Fig. 6a, please indicate the positions of the S1 and S2 sites and the UBD.

Interacting residues of the S1, S2 and UBD have been highlighted in new Fig 7a.

10. In Fig.6b and Supplementary Fig. 1c, the cleavage of K48-linked Ub chains by O. chuto DUB (1-299) and O. tsutsu DUB (1-177) should be tested.

The original Figure 6b has been replaced with Figure 7d, which directly compares the OtDUB 1-177 and 1-311 polypeptides to OcDUB 1-777 and 1-299 for both K63 and K48 extended chain cleavage.

11. Phe59 of OtDUB is replaced by Thr in *O. chuto* DUB (Fig. 6a). In Fig. 3c, the F59T mutation of OtDUB decreases the cleavage activity for K63-linked chains. However, in Fig. 6b, *O. chuto* DUB shows substantial activity for K63-linked chains. Does *O. chuto* DUB really have the S2 site? Please explain this.

Originally, we were also wondering if the OtDUB F59T would make K63 cleavage comparable to OcDUB, but to our surprise, the mutation slowed cleavage down, suggesting the S2 site normally assists in K63 chain cleavage and is no longer functional in OtDUB F59T. It was an excellent idea to test if OcDUB has an S2 site at all, which would clarify the F59T OtDUB result. To this end we performed titration experiments with increasing molar equivalents of ubiquitin incubated with either OtDUB₁₋₁₇₇-C135A or OcDUB₁₋₁₇₇-C134A. To simplify interpretation, we removed the UBD and mutated the active site for stable interaction with ubiquitin. In these titration experiments, shown in new Figure 7f and Supplementary Figure 6a, the OtDUB DUB domain clearly has the capacity to bind 2 ubiquitins (via the S1 and S2 sites), while the OcDUB appears to only bind ubiquitin in a 1:1 ratio, arguing against an S2 site in OcDUB. Furthermore, OtDUB₁₋₁₇₇-F59T,C135A mirrored the OcDUB₁₋₁₇₇-C134A data suggesting the F59T mutation is sufficient for disrupting the S2-site.

12. Is pre-incubation performed in Fig. S1b also effective for UBD mutants (V203D, F207D and L221D)?

The short answer is no. For extended chain cleavage where the OtDUB is at 500 nM (a concentration where self-inhibition is strongest), the V203D mutant actually alleviates the self-inhibition and results in complete K48/K63 substrate cleavage irrespective of ubiquitin preincubation (Figure 6h, Supplementary Figure 5f). For extended chain cleavage with 50 nM OtDUB (where self-inhibition is insignificant), preincubation with ubiquitin has no added benefit (data not shown). For di-ubiquitin cleavage, preincubation of V203D with ubiquitin had no added effect on cleavage for K48 or K63 (Figure 6a, quantified in 6b and Supplementary 5a). We think the major reason we see higher activity for V203D compared to WT in all of these cases is the UBD has such a high affinity it normally sequesters substrate from the active site. If one saturates the UBD by preincubation with ubiquitin, then extremely efficient cleavage of K48 di-ubiquitin is observed (Figure 6a, b).

13. In Fig. S2, please compare OtDUB-Ub with OTUB1-Ub, where two Ub molecules are aligned so as to form a K48 linkage.

Thank you for suggesting this comparison. This alignment is illustrated below. OTUB1-Ub complex structure is shown on the left, with OTUB1 in green cartoon, S1 Ub in grey cartoon, and S2 Ub in red surface. Alignment of S1 Ub's between the OTUB1-Ub and OtDUB-Ub structure is shown on the right, with OtDUB in cyan, VR-1 in slate, and only Ub molecules from the OTUB1-Ub structure visible (same coloring as left). This alignment clearly reveals severe clashes with the distal, K48-linked S2 Ub (red surface) from OTUB1-Ub, demonstrating that the binding mode OTUB1 utilizes to stabilize a di-K48-linked chain cannot be employed by OtDUB.

14. In Methods, twin refinement is mentioned. Twinning parameters should be shown.

Twinning parameters have been added.

Minor points:

1. The terms "OTT_1962" and "OtDUB" are used in the figures. Please use either one.

OTT_1962 has been replaced with OtDUB throughout the figures and text.

2. In Fig. 2, the position of Lys48 should be indicated.

Lys48 has been indicated

3. "Supplemental Figure" may be "Supplementary Figures" in Nature style.

Corrected.

4. In Methods, "DNAse" should be "DNase".

Corrected.

Reviewer #2 (Remarks to the Author):

In their submitted manuscript, Berk & Lim et al. present a thorough structural and biochemical analysis of a novel deubiquitinase domain from the scrub typhus pathogen *Orientia tsutsugamushi*. This work is a fascinating addition to our understanding of a growing family of bacterial deubiquitinases from the CE clan of proteases and describes several discoveries that are thus far unique to OtDUB and related *Orientia* homologs. In particular, the preference for longer polyubiquitin chains and surprising high-affinity interaction with monomeric ubiquitin are very striking and new to this family of deubiquitinases. The work is of high quality and will be of broad interest to researchers studying the ubiquitin system or host-pathogen interactions. Barring several comments outlined below, I find this manuscript to be well-suited for publication in *Nature Communications*.

1) The authors seem to explain the lack of activity against di-ubiquitin chains as resulting from a requirement for longer (K63-linked) chains binding into the S2 site. If this is true, then why is OtDUB also more active against longer K48-linked chains, when based on the structure the S1-S2 connectivity appears to be K63-specific?

First, we have now repeated the di-ubiquitin cleavage assays found that certain di-ubiquitins, including K48 and K63-linked ones, are in fact cleaved, albeit at slower rates than longer chains (trimers and tetramers). Second, without supporting structural supporting data, we can only speculate on the enhanced cleavage of longer K48 chains: one possibility is K48 chains also sample the S2 site with the distal ubiquitin making different contacts than the K63 distal ubiquitin. This idea is in line with the biochemistry for K48 chain cleavage and the F59T S2-site mutation. K48 tetra-ubiquitin chain cleavage is slightly impaired at early time points by the F59T mutation (Figure 3c, d).

2) On a related note, why is it that K48 chains can be cleaved to produce monoUb, while K63 chains cannot (see Fig. 1e-f)? And why is the enzyme active against the simple Ub-AMC substrate?

Thank you for this observation; it had puzzled us (and the other reviewers) too (see response to Reviewer #1 (points 4, 8). Given the clear activity against Ub-AMC and the

reviewer comments, we decided to retest diUb cleavage by first testing the commercial diUb panel with SYPRO Ruby staining rather than anti-ubiquitin blotting (new Figure 1e). With this readout, cleavage of both K48 and K63 diUb was detectable, and we reexamined all our mutants with K63 and K48 diUb. Although we do not fully understand all the original puzzling data, we suspect that during the Western transfer process, much of the mono-Ub was being transferred through the PVDF membranes (something that is known to occur), and we were just on the edge of detection for mono-Ub in these blots. We have now replaced the blotting results with the gel stain readouts throughout.

3) Considering the above comments, I am curious if the active site (S1'-S1 connection) is incompatible with K63-linked chains, while other linkages (e.g. K48 or Ub-AMC) can be cleaved. Have the authors considered the possibility that OtDUB targets hybrid chains?

The S1'-S1 connection may be incompatible to some degree for either K48 or K63 chains, but this is difficult to speculate on without structural information about the S1' site. Hybrid chain cleavage is a possibility, especially given the promiscuity of the DUB against K11,33,48, and 63 di-ubiquitin chain types. This is an attractive question for future studies.

4) In Figure 4, the authors present mutations along one face of VR-1 that target the S1 site. These mutations appear to affect cleavage of K48 chains but not K63 chains, presumably because the effect on K63-linked chains is masked by the additional S2 interaction. Can the authors support this with a Ub-AMC assay? And possibly with S1-site mutations in the VR-2 region as well?

As suggested, we have applied the Ub-AMC assay to look at activity of the S2 site mutant F59T (new Figure 3e). The Ub-AMC assay reveals activity equal to WT OtDUB 1-259, as expected. K63 diUb cleavage, but not that of K48 diUb, is accelerated by the F59T mutation (Figure 3f, g), as predicted if this helps prevent diUb displacement from the active site by S2 binding. Furthermore, when F59T is preincubated with ubiquitin (Fig 6e and f) the cleavage of K63-linked dimers is still accelerated, suggesting ubiquitin binding to another site is required for maximal turnover; this binding is likely to be to the UBD, which would allow full folding of the domain, including the VR-1 surface that contributes to S1.

5) Given the disorder-order transition observed in VR-1 upon Ub binding (Fig. 5f), I am surprised that blocking this interaction with the V203D mutation does not maintain the disordered state enough to affect Ub binding to the S1 site and thus DUB activity (Fig. S4a). By eye this mutant might be slightly slower than the WT enzyme. Quantitation of this assay and/or kinetic analysis with the Ub-AMC substrate could be more telling and possibly indicate a role for the OtDUB VR-1 UBD in activating DUB function (akin to the interesting regulation of OTUB1 in humans).

The new Figure 6 and Supplementary Figure 5 of the revised manuscript contains a comprehensive analysis of the V203D mutant by Ub-AMC, di-ubiquitin and tetra-ubiquitin substrate analyses. K63 di-ubiquitin (Figure 6a) and tetra-ubiquitin cleavage (Supplementary Figure 5d, e [old Fig. S4a and accompanying quantification]) both show reduced cleavage. The Ub-AMC assay reveals reduced activity of V203D, equivalent to OtDUB₁₋₁₇₇. At high enzyme concentrations the V203D acts similar to OtDUB₁₋₁₇₇ as well where both can cleave more efficiently than OtDUB₁₋₂₅₉ due to a loss of the negative regulating UBD (Figure 6h, Supplementary Figure 5f).

Minor comments:

1) The OcDUB appears to be more active against di-ubiquitin substrates. Have the authors considered performing the full di-ubiquitin cleavage panel with this enzyme to profile specificity across the active site?

OcDUB₁₋₂₉₉ exhibits specificity comparable to OtDUB₁₋₂₅₉, but is more efficient at cleaving the targeted linkage-types. New Figure 7c.

2) Without experimentation, claims that the UBD is specific for mono-Ub over chains are perhaps overstated.

This was incorrectly phrased. We adjusted the text as follows (p. 12):

Our data indicate the UBD binds mono-ubiquitin and likely does not prefer particular ubiquitin chain types.

3) In future assays, the “time zero” samples would be more representative if the individual reaction components were mixed directly into SDS sample buffer.

Thank you for this suggestion for consideration in future studies.

4) The authors might consider discussing the tUI ubiquitin sensor published by the Cohen lab as a higher affinity (though artificial) ubiquitin binder.

A sentence highlighting this work was included in the discussion. “Related to this, avidity-based, high-affinity protein binders specific for free ubiquitin have been engineered that are useful as sensors for free mono-ubiquitin levels⁵³.”

5) Please include CC1/2 values in Table 1, or otherwise justify why such an aggressive I/sigI cutoff was acceptable.

CC1/2 values have been added to Table 1.

6) Why did the authors choose to not use the higher resolution native dataset for apo OtDUB?

Although the native dataset diffracted to higher nominal resolution, these crystals adopted a thin plate crystal habit which led to severe anisotropy and an overall decrease in completeness and redundancy statistics. For the SeMet derivatized crystals, crystallization conditions were optimized to improve overall dataset quality.

Sincerely,

Jonathan Pruneda

Reviewer #3 (Remarks to the Author):

Berk et al. reported the structural characterization and ubiquitin chain linkage specificity of a deubiquitylase from *Orientia tsutsugamushi*. The crystal structures of the DUB domain of OtDUB alone and bound to three molecules of ubiquitin were presented. The structure of OtDUB reveals a similar core fold of CE clan proteases with distinct variable regions that are involved in ubiquitin binding. Surprisingly when co-crystallizing OtDUB with free ubiquitin, there were three ubiquitin bound in the structure. One bound at the S1 site, one at the S2 site and the other bound to a unique backside surface of the C-terminal region of OtDUB which the authors referred to as the ubiquitin-binding domain (UBD). Interestingly the C-terminus of ubiquitin at S2 is in proximity of K63 of ubiquitin at S1 leading to the hypothesis of K63-ubiquitin chain ($n > 3$) recognition. The authors showed that OtDUB has preference for cleaving longer K63 ubiquitin chain ($n > 2$) and does not cleave di-ubiquitin.

Likewise it also cleaves K48-ubiquitin chain. The structure explains a model for how OtDUB could bind a tri-K63-ubiquitin. Mutation in the S2 binding site reduced cleavage of K63-tetra-ubiquitin but had little effect on K48-tetra-ubiquitin. While it is unclear how OtDUB recognizes K48-ubiquitin chain, the structure showed that K48 of S1 ubiquitin is accessible for conjugation with ubiquitin. Lastly the authors showed that the backside of UBD has a strong binding affinity for ubiquitin (K_d of 5 nM), which is two-orders of magnitude tighter than previously reported UBD-ubiquitin interaction.

This work reports a novel structure of OtDUB in complex with ubiquitin at S1 and S2 sites and explains how it recognizes K63-ubiquitin chain. In addition, it identifies a high-affinity

UBD. These findings will be of interests to researchers in the ubiquitin field in understanding ubiquitin recognition and chain specificity. There are few concerns see below.

Comments

1. It is a bit confusing that in some assays 1-259 (Figure 3c, 4b, 6b, S4) was used and in others 1-311 was used (Figure 1d-f, S1B). Do both constructs have similar activity? It seems that 1-259 was used when comparing with mutants. Please explain.

When our analyses started we were working with 1-311. The crystal structure of 1-311 only resolved residues 1-259, so we figured 260-311 were less structured and switched to 1-259. By Ub-AMC cleavage, 1-259 does exhibit slightly faster cleavage kinetics (new Supplementary Figure 2b). For Ub chain cleavage, 1-311 and 1-259 exhibited comparable activities against K48 and K63 diUb at 50 nM and 500 nM concentrations of OtDUB (Supplementary Figures 2c). For K48 and K63 tetraUb, they were equally active at 50 nM, but 1-259 was slightly faster at 500 nM due to the self-inhibitory effects at this concentration (Supplementary Figure 2a). To reduce confusion, all 1-311 activity assays were moved to the supplementary material.

2. This reviewer assumes that the same amount of OtDUB was used in the assay for WT and mutant. Please show a SDS-PAGE of OtDUB WT and mutants used in this study to illustrate their purity.

New Supplementary Figure 6c shows a Coomassie stained gel demonstrating the purity of enzymatic fragments and mutants used in activity assays.

3. In the supplementary text and Supplementary Figure 1B, the authors showed that in the absence of added free ubiquitin, OtDUB failed to cleave K63-tri-ubiquitin completely due to product inhibition. Is free ubiquitin added in all assays in other figures? It seems that K63-tetra-ubiquitin (Fig. 3b, 4b versus Supplementary Figure 1B) is less susceptible to inhibition (by 10 min nearly all tetra-ubiquitin is cleaved). Is this due to tri vs tetra-ubiquitin? Is this specific to K63 or the same for K48-ubiquitin chain?

Unless noted otherwise, all cleavage assays were performed in the absence of preincubation with ubiquitin. The experiment in Supplementary Figure 1b was done with 500 nM OtDUB; in experiments where extended chains, which are cleaved quickly, the DUB concentration was at 50 nM. This alone was sufficient to show increased activity in

vitro, as shown directly in new Supplementary Figure 2a for both K63 and K48 tetraUb chains. The self-inhibition at higher concentrations was also true for K48 triUb chains (data not shown). This self-inhibition is attributed to the UBD. When the UBD is mutated (V203D) extended chain cleavage with 500 nM of OtDUB is now highly efficient (Figure 6h, Supplementary Figure 5f). Related to this, the preincubation of the OtDUB with ubiquitin at 500 nM also accelerates extended chain cleavage (Figure 6h, Supplementary Figure 5f).

The authors stated in the supplementary text that 1-177 (Fig. S1B) is more efficient in cleaving K63-tetra-ubiquitin compared to 1-311 (Fig. 1E) suggesting that UBD could compete for ubiquitin binding thereby inhibiting the cleavage. First, these reactions were performed on separate western blots and it should be performed on the same gel to make such comparison. Second, Fig S4A showed that V203D abrogated UBD-ubiquitin interaction and exhibited similar activity as WT. If UBD has an inhibitory role one would expect V203D to be more active. Lastly, Fig.4A right panel showed that VR-1 makes extensive contact with S1 ubiquitin and 1-177 construct lacks this whole surface. One would predict to have a more severe defect than the point mutations in Fig. 4B and could impact on the cleavage of K63 chain. Please explain.

In light of our new data showing OtDUB can in fact cleave diUb based on SYPRO Ruby gel staining, Figure 1e, f and Supplementary Figure 1b were removed. In their place Supplementary Figure 2a shows 1-177 is more efficient at cleaving than 1-311 at the 500 nM OtDUB concentration for both K48 and K63 tetraUb chains.

There are instances where we see increased activity for the V203D mutant: against K63 tetraUb (Figure 6h), K48 tetraUb (Supplementary Figure 5f) and K48 diUb (Figure 6a, b).

The consequences of lacking the VR-1 S1 site contacts in 1-177 has variable effects dependent on chain length and OtDUB concentration. For both K48 and K63 diUb and Ub-AMC, kinetics are slower with 1-177 than the longer fragments, at both low and high OtDUB concentration (Supplementary Figures 2b, c). The impact here on diUb is the most telling for the requirement of a fully intact VR-1 for proper cleavage. For extended chains (K48 and K63) the cleavage by 1-177 is *more* efficient than 1-259 and 1-311 when the OtDUB is at 500 nM due to the self-inhibitory defects of the UBD (Supplementary Figures 2a). At 50 nM of OtDUB, the cleavage of K63 extended chains by 1-177 is comparable to 1-259 and 1-311 (Supplementary Figure 2a), but is reduced for K48 extended chains. The likely explanation for this observation is the S2 site prefers K63 chains (based on the complex structure) and this positions extended K63 chains more efficiently into the active site even when half of the S1 site is absent.

It would be useful to speculate the role of UBD-ubiquitin interaction in the discussion.

We included the following text in the discussion; “Interestingly, OtDUB_{UBD}:ubiquitin interaction strongly promotes DUB activity at high OtDUB concentrations. Saturation of the UBD with free ubiquitin both reduces competition for ubiquitylated substrates at the DUB active site and enhances S1-site formation by promoting folding of VR-1/UBD, which increases K48-chain recognition. Local subcellular concentrations of OtDUB during *Orientia* infection are not yet known, so it is unclear how relevant this is during infection. The high cytoplasmic levels of free ubiquitin may guarantee a fully activated DUB.”

4. In Figure 3C, it seems that K63-tetra-ubiquitin is cleaved to di-ubiquitin whereas K48-tetra-ubiquitin is cleaved to mono-ubiquitin. Please discuss this difference. Also in Figure S1B, K48-di-ubiquitin was not cleaved at all, but when tetra-ubiquitin was used even di-ubiquitin was cleaved. Please explain.

In Figure 3c, WT OtDUB stalls at diUb when using K63 tetraUb. This is most likely due to the presence of the S2 site having a preference for K63 chains –as inferred from the Ub-OtDUB structure– and this sequesters the K63 diUb from the active site. If the time points were extended in this figure, one would likely see a reduction in K63 diUb levels for F59T, which has an inactivated S2, as is seen in Figure 3f and 6e.

Supplementary Figure 1b was assayed by western blot which may have prevented detection of diUb cleavage. Related to this we do see some variability in the cleavage kinetics for K48 diUb, with very little visible cleavage occurring in certain experiments (Figures 4e, 6c, 6e) while other times it is more efficient (Figures 1e, 3f, 6a, Supplementary Figure 2c). The basis for the differences here is unclear as the enzyme is still obviously active because cleavage is very efficient when preincubated with ubiquitin in the same experiments for Figures 6c, 6e.

5. Figure 1D legend missing. Please specify the concentration of Ubl-AMC or 1X

Thank you for catching this error. The figure legend was adjusted accordingly.

6. p.16 top “Instead the backside of the UBD domain (VR-1) is necessary for optimal K48-chain cleavage (Figure 4B and 4C)”. Throughout the manuscript backside was referred to UBD-ubiquitin interaction. Please revise to S1 binding site.

This specific sentence has been removed from the discussion during the revision process. The text now consistently refers to the backside of VR-1 as the UBD and not vice versa. We continue to refer to this domain as the VR-1 as it represents a small fraction of the S1 site.

REVIEWERS' COMMENTS:

Reviewer #1 (Remarks to the Author):

NCOMMS-19-37565A

Review comment:

Berk et al., "A Deubiquitylase with an Unusually High-affinity Ubiquitin-binding Domain from the Scrub Typhus Pathogen *Orientia tsutsugamushi*"

All my concerns have been properly addressed in the revised manuscript. However, I raise some points for improvement of terminologies as follows:

1. Alphabets indicating coefficients, constants and space groups are traditionally italicized (e.g. 'K'*d*, 'CC'*1/2*, 'P'*21212*, mean '*l*', and so on).
2. In Table 1, a Greek alphabet should be used for "sigma". Percentage representation is not suitable for CC*1/2*. The unit of B factors (i.e., Å²) should be presented.
3. In the Methods section, the name of the column for gel-filtration chromatography 'S75' is not accurate, and should be corrected to 'Superdex 75'.
4. The isolated OtDUB domains are termed 'polypeptides' in some sections and 'OtDUB fragments' in other sections. Please use either one. I favor the latter.
5. In the legend for Supplementary Figure 6, '... for enzymatic analyzes were ...' should be corrected to '... for enzymatic analyses were ...'.

Reviewer #2 (Remarks to the Author):

Berk and Lim et al. have made significant improvements to their detailed analysis of the OtDUB Ub binding sites and their roles in regulating and specifying DUB activity. My comments have been nicely addressed and I only have two minor points for the authors' consideration.

1) To explain the enhanced cleavage of longer K48 chains, instead of a VR-2:UbS2 interaction that would be rotated compared to their structure, could there instead be an S2' site that dictates K48 specificity between the S1' and S2' positions? If so, one would predict that DUB activity toward long K48 chains would be "exo" in nature whereas K63 cleavage would be "endo", and this might explain the observation in Fig 3C that K63 cleavage stalls at diUb while K48 cleavage releases monoUb.

2) It's possible that I missed it, but I could not find a reference to Fig 2C in the text.

Reviewer #3 (Remarks to the Author):

The authors have addressed all my concerns and provided additional data to support their findings. The manuscript is suitable for publication.

Minor comments

1. Figure 6f comes before Figure 6e in the text.
2. Typo "Abode Photoshop" in the methods

REVIEWERS' COMMENTS NCOMMS-19-37565A:

Reviewer #1 (Remarks to the Author):

Review comment:

Berk et al., "A Deubiquitylase with an Unusually High-affinity Ubiquitin-binding Domain from the Scrub Typhus Pathogen *Orientia tsutsugamushi*"

All my concerns have been properly addressed in the revised manuscript. However, I raise some points for improvement of terminologies as follows:

1. Alphabets indicating coefficients, constants and space groups are traditionally italicized (e.g. 'K'd, 'CC'1/2, 'P'21212, mean 'l', and so on).

Formatting has been corrected according to the reviewer's suggestions and in line with Nature journal guidelines.

2. In Table 1, a Greek alphabet should be used for "sigma". Percentage representation is not suitable for CC1/2. The unit of B factors (i.e., Å²) should be presented.

Table 1 has been formatted according to Nature guidelines and the reviewer's suggestions.

3. In the Methods section, the name of the column for gel-filtration chromatography 'S75' is not accurate, and should be corrected to 'Superdex 75'.

S75 was replaced with Superdex 75 throughout the text.

4. The isolated OtDUB domains are termed 'polypeptides' in some sections and 'OtDUB fragments' in other sections. Please use either one. I favor the latter.

Polypeptides were replaced with fragments

5. In the legend for Supplementary Figure 6, '... for enzymatic analyzes were ...' should be corrected to '... for enzymatic analyses were ...'.

Corrected.

Reviewer #2 (Remarks to the Author):

Berk and Lim et al. have made significant improvements to their detailed analysis of the OtDUB Ub binding sites and their roles in regulating and specifying DUB activity. My comments have been nicely addressed and I only have two minor points for the authors' consideration.

1) To explain the enhanced cleavage of longer K48 chains, instead of a VR-2:UbS2 interaction that would be rotated compared to their structure, could there instead be an S2' site that dictates K48 specificity

between the S1' and S2' positions? If so, one would predict that DUB activity toward long K48 chains would be "exo" in nature whereas K63 cleavage would be "endo", and this might explain the observation in Fig 3C that K63 cleavage stalls at diUb while K48 cleavage releases monoUb.

An S2' site that dictates K48 chains cannot be ruled out and would be an exciting twist to the OtDUB mechanism. It will be interesting to directly test OtDUB against distinct chain substrates in future work to determine if it processes chains differently (exo vs endo).

The residual K63 di-Ub in Fig 3c is likely due to it binding in the S2-S1 position, which out competes the S1-S1' position/cleavage. This is supported by the accelerated cleavage of K63 di-Ub with the F59T mutation (Fig 3f,g).

2) It's possible that I missed it, but I could not find a reference to Fig 2C in the text.

Added a sentence describing the rotation of UbS1 in OtDUB relative to other CE clan DUBs.

Reviewer #3 (Remarks to the Author):

The authors have addressed all my concerns and provided additional data to support their findings. The manuscript is suitable for publication.

Minor comments

1. Figure 6f comes before Figure 6e in the text.

Given the figure layout it would be difficult to reorder the images and we would prefer to keep 6f before 6e in the text. We have written "(see Figure 6f)" in the text on p. 14 as is conventionally done to indicate a figure reference out of order with a figure or figure panel.

2. Typo "Abode Photoshop" in the methods

Corrected. Thanks for catching!